# Activation of Intact Bacteria and Bacterial Fragments Mixed with Agar as Cloud Droplets and Ice Crystals in Cloud Chamber Experiments

Kaitlyn J. Suski,[1] David M. Bell,[1,2] Naruki Hiranuma,[3,4] Ottmar Möhler,[3] Dan Imre,[5] and Alla Zelenyuk[1]

[1]Pacific Northwest National Laboratory, Richland, WA, USA.
[2]Currently at Paul Scherrer Institute, Villigen, Switzerland.
[3]Institute of Meteorology and Climate Research – Atmospheric Aerosol Research, Karlsruhe Institute of Technology, Karlsruhe, Germany.
[4]Currently at Department of Life, Earth and Environmental Sciences, West Texas A&M University, Canyon, TX, USA.
[5]Imre Consulting, Richland, WA, USA

*Correspondence to:* Alla Zelenyuk (alla.zelenyuk-imre@pnnl.gov)

**Abstract.** Biological particles, including bacteria and bacterial fragments, have been of much interest due to the special ability of some to nucleate ice at modestly supercooled temperatures. This paper presents results from a recent study conducted on two strains of cultivated bacteria which suggest that bacterial fragments mixed with agar, and not whole bacterial cells, serve as cloud condensation nuclei (CCN). Due to the absence of whole bacteria cells in droplets, they are unable to serve as ice nucleating particles (INPs) in the immersion mode under the experimental conditions. Experiments were conducted at the Aerosol Interaction and Dynamics in the Atmosphere (AIDA) cloud chamber at the Karlsruhe Institute of Technology (KIT) by injecting bacteria-containing aerosol samples into the cloud chamber and inducing cloud formation by expansion over a temperature range of -5 to -12 °C. Cloud droplets and ice crystals were sampled through a pumped counterflow virtual impactor inlet (PCVI) and their residuals were characterized with a single particle mass spectrometer (miniSPLAT). The size distribution of the overall aerosol was bimodal, with a large particle mode composed of intact bacteria and a mode of smaller particles composed of bacterial fragments mixed with agar that were present in higher concentrations. Results from three expansions with two bacterial strains indicate that the cloud droplet residuals had virtually the same size distribution as the smaller particle size mode and had mass spectra that closely matched those of bacterial fragments mixed with agar. The characterization of ice residuals that were sampled through an ice-selecting PCVI (IS-PCVI) also shows that the same particles that activate to form cloud droplets, bacteria fragments mixed with agar, were the only particle type observed in ice residuals. These results indicate that the unavoidable presence of agar or other growth media in ***all*** laboratory studies conducted on cultivated bacteria can greatly affect the results and needs to be considered when interpreting CCN and IN activation data.

# 1 Introduction

Aerosols affect climate directly, by scattering and absorbing solar and terrestrial radiation and indirectly through interactions with clouds and precipitation (IPCC, 2013). In their role as cloud condensation nuclei (CCN) and ice nucleating particles (INPs), aerosol particles affect cloud microphysical properties (Pruppacher and Klett, 2010). Biological particles, including
bacteria, which are found in the atmosphere in various forms but at relatively low number concentrations (Despres et al., 2012), might play an important role globally as CCN or INPs (Morris et al., 2004;Möhler et al., 2007). While a large body of work on the ice nucleating (IN) activity of biological particles exists, the extent of their role in ice formation within modestly supercooled clouds remains poorly constrained (Murray et al., 2012;DeMott and Prenni, 2010;Möhler et al., 2007;Hoose et al., 2010;Spracklen and Heald, 2014;Phillips et al., 2009).

Previous studies that looked at the CCN activity of bacteria concluded that bacteria range from very hygroscopic, and hence CCN active under atmospherically relevant conditions (Bauer et al., 2003), to only slightly hygroscopic, activating only at critical supersaturations close to or exceeding those indicating wettability (Franc and DeMott, 1998). It has been suggested, based on contact angle measurements (Sharma and Rao, 2002), that these differences reflect a range of bacteria cell wall types, some of which are hydrophilic, while others are hydrophobic. However, most of these studies utilized aerosolized suspensions
of cultured cells that include, in addition to intact bacteria with well-defined cell walls, bacterial fragments mixed with agar growth medium, in which the bacteria were cultured (Wolf et al., 2015). It has been suggested that biological particles may serve as giant CCN (Möhler et al., 2007) and because bacterial samples have been shown to induce ice formation at relatively warm temperatures it has been suggested that they might play an important role as INPs in mixed phase clouds (Möhler et al., 2007).

Depending on the INP composition and size, and on the temperature and relative humidity (RH), ice formation can occur by several pathways. The freezing modes most relevant to bacteria particles are *immersion freezing*, which is induced by a particle immersed in supercooled water (Murray et al., 2012;Vali et al., 2015;Vali et al., 1978); *condensation freezing*, in which water condensing on an INP freezes (Pruppacher and Klett, 2010); and *contact freezing* that occurs when a liquid droplet hits an INP and freezes on contact (Cantrell and Heymsfield, 2005). However, for the experiments presented here immersion freezing
represents the dominant ice formation mechanism (Gallavardin et al., 2008). In immersion freezing, the INP first activates as a liquid cloud droplet, when the RH over water ($RH_w$) exceeds 100% RH, and subsequently ice forms, which means that this ice formation mechanism is tightly connected to the particle CCN activity (Murray et al., 2012;Vali et al., 2015;Vali et al., 1978).

Many biological particle types including pollen (Hader et al., 2014;Augustin et al., 2013;Diehl et al., 2001), fungi (Fröhlich-Nowoisky et al., 2015), and decaying plant material (Schnell and Vali, 1976;Conen et al., 2016) have been identified as
efficient INPs. Additionally, IN active macromolecules have been observed from fungi (O'Sullivan et al., 2016), pollen (Pummer et al., 2012), and proteins from IN active bacteria (Wolber et al., 1986). These IN active proteins were shown to be active on their own, in the absence of intact bacteria, demonstrating that a single protein is all that is required to initiate freezing

(Govindarajan and Lindow, 1988). Additionally, IN active bacterial fragments have been observed in the ambient environment (Šantl-Temkiv et al., 2015), mineral dust has been observed to be co-located with biological residues in mixed phase clouds (Creamean et al., 2013), and IN active proteins from fungi have been shown to attach to dust (O'Sullivan et al., 2016), which can be used as a carrier to transport these biological macromolecules.

Pseudomonas syringae is one bacteria strain that has been shown to nucleate ice at temperatures as high as -2 °C (Maki et al., 1974;Vali et al., 1976). This bacteria's special ability to nucleate ice at relatively warm temperatures has been shown to cause frost injury to plants (Lindow et al., 1982) leading to decades long speculation that it may play a role in precipitation processes (Vali et al., 1976). A recent study showed that Pseudomonas syringae has a unique structure that orders water at its surface through hydrogen bonding, which allows for efficient latent heat removal, thus enhancing ice formation (Pandey et al., 2016).

A store-bought snow inducer, Snomax®, which contains Pseudomonas syringae has been used in laboratory ice nucleation studies. A recent study shows that the ice nucleation ability of Snomax® changes based on sample age and after repeated freeze and thaw cycles (Polen et al., 2016). A different strain of Pseudomonas bacteria, Pseudomonas Fluorescence (PF CGina), that was isolated from glacier melt water in Antarctica was also shown to nucleate ice at warm temperatures (Attard et al., 2012;Amato et al., 2015). Pseudomonas syringae and PF CGina were shown to have decreased viability when exposed

to UV light, but only significantly decreased IN activity when exposed to acidic conditions, suggesting that the denaturation of proteins and not cell death has the largest effect on the IN activity of these bacteria (Attard et al., 2012). It was confirmed that cell death does not significantly reduce IN activity of Pseudomonas syringae and PF CGina by Amato et al. (2015).

Several laboratory studies have identified proteins and protein complexes as the ice nucleating entities in Pseudomonas syringae (Lindow et al., 1989;Maki and Willoughby, 1978;Hartmann et al., 2013;Govindarajan and Lindow, 1988).

Additionally, 3 protein classes were identified based on their abundance and different ice nucleating abilities (Turner et al., 1990) and it was shown that non-protein compounds that anchor proteins to cell membranes were present in the most IN active classes (Turner et al., 1991). A previous study at the AIDA cloud chamber concluded that both bacterial fragments and whole cells were CCN and IN active, with whole cells being slightly more IN active than the fragments, and the fragments having a higher hygroscopicity than the whole cells (Oehm, 2012). Several other studies have reached similar conclusions that both

whole cells and bacterial fragments serve as INPs (Wex et al., 2015;Hartmann et al., 2013;Yankofsky et al., 1981). In contrast, an earlier study at the AIDA chamber suggested that only intact bacteria are IN active (Möhler et al., 2008). The distinction between whole cells and bacterial fragments is important from many perspectives, especially when calculating available surface area, surface site density, and active or frozen fraction of bacteria in laboratory experiments. More importantly, modeling studies suggest biological particles may be an important player in precipitation processes (Burrows et al.,

2013;Morris et al., 2004), but because atmospheric concentrations of intact bacteria and bacterial fragments are poorly constrained and expected to be vastly different, it remains uncertain how big of a role bacteria play in mixed phase cloud formation globally.

When interpreting previously published studies, it is important to keep in mind that agar or other growth media are present in *all* laboratory studies that are done with cultured bacteria. An aerosolized solution of cultivated bacterial cells yields at least two particles types: intact or whole bacterial cells and smaller particles, composed of bacterial fragments mixed with agar. The size distribution of the small particle mode depends on the particle generation, i.e. the size of aerosolized droplets, and the concentration of the agar and bacterial fragments in the solution/suspension. Therefore, it would not be surprising to find that the particle number concentration and size distribution differ from one experimental set-up to the next, and that it could even change with the age of the solution/suspension. Most importantly, in order to use laboratory studies to generate useful parameters for atmospheric models one needs to understand the role agar plays in the CCN and IN activity of bacteria.

The data presented here were acquired during cloud simulation experiments performed at the AIDA cloud chamber during the first part of the Fifth International Ice Nucleation Workshop (FIN-1). The single particle mass spectrometer, miniSPLAT (Zelenyuk et al., 2015) was utilized to characterize the size and chemical compositions of intact bacteria cells and bacterial fragments mixed with agar before and after cloud formation. Most importantly, it was used to characterize the residuals of liquid droplets and ice crystals formed during the expansions. Cloud particles were separated from interstitial particles with a pumped counterflow virtual impactor (PCVI) or an ice-selecting PCVI (IS-PCVI) (Hiranuma et al., 2016). Comparison between the properties of cloud residuals with the overall particle population in the chamber before and after the expansion allows for the determination of cloud and ice active versus interstitial particles. As it will be demonstrated below, the results of these measurements, on two different bacteria, indicate that bacterial fragments mixed with agar accounted for the clear majority of particles active as CCN and as INPs, while whole cells activated with much lower probability, if at all.

## 2 Experimental

The Fifth International Ice Nucleation Workshop (FIN) was a three-part study that aimed to compare a number of single particle mass spectrometers and ice nucleation instruments. The data presented here were collected during FIN-1, which took place in November of 2014 at the AIDA cloud chamber at KIT in Karlsruhe, Germany. Ten single particle mass spectrometers from several research groups around the world were brought together for FIN-1 to simultaneously characterize the size and composition of various types of aerosol particles, including those being used in the AIDA chamber to study their activity as CCN and INPs, Various aerosol types were sampled from the AIDA cloud chamber before, during, and after expansions forming cloud particles and each of these mass spectrometers sampled using their own protocol. The results of the intercomparison will be the subject of future publication. The present manuscript presents the results of measurements, made by miniSPLAT only, during three expansions using two bacteria.

The AIDA cloud chamber and its operation were previously described in detail elsewhere (Möhler et al., 2003). Briefly, the AIDA cloud chamber is an 84 $m^3$ aluminum vessel in a thermally insulated housing that can be set to a desired RH, temperature, and pressure. To induce cloud formation, the pressure of the chamber, filled with aerosol particles, is lowered by pumping on

the chamber, which causes the gas to expand, the temperature to drop, and the RH to increase above ice saturation, depending on the specific experiment. Once supersaturation with respect to water is reached at temperatures below 0°C, cloud droplets, and eventually ice crystals, form. The data presented in this paper are from 3 expansions during the campaign, Expansions 36, 37, and 38, which are referred to as Expansions 1, 2, and 3, respectively, throughout this paper. The mobility size distributions

of particles in the AIDA chamber are measured before and after expansions with a scanning mobility particle sizer (SMPS), comprised of a differential mobility analyzer (DMA, TSI Inc., Model 3080) and a condensation particle counter (CPC, TSI Inc., Model 3010). The SMPS sizes particles with mobility diameters larger than 14 nm and smaller than 740 nm, while larger particles, with aerodynamic diameters from 0.5 to 20 µm, are sized with an aerodynamic particle sizer (APS, TSI Inc., Model 3321). Throughout the manuscript the combined SMPS and APS data are shown as volume-equivalent diameters calculated

using a particle density of 1.4 g/cm$^3$ based on previous measurements (Wex et al., 2015) and optimized assuming a dynamic shape factor of 1 according to Peters et al. (2006). A similar application of volume equivalent diameter in SMPS and APS for use in conjunction with AIDA data can be found in Wex et al. (2015). Total particle number concentrations are measured with a CPC (TSI Inc., Model 3076). Temperature and RH are measured using a number of calibrated sensors (Fahey et al., 2014;Möhler et al., 2003). Optical diameters of cloud droplets and ice crystals are determined using two optical particle

counters (welas OPCs, Palas GmbH, sensor series 2300 and 2500). The detection size ranges of the two OPCs, referred to as welas1 and welas2, are ~0.7 to 46 µm and ~5 to 240 µm, respectively (Wagner and Möhler, 2013). The simultaneous use of these two OPCs effectively spans the size range of droplets and ice crystals formed in the chamber. A rough estimate of the number of particles that activated as CCN (CCN/CN) can be calculated by dividing the CPC measured particle concentration prior to the expansion by the maximum value of welas2 measured cloud particle concentrations observed during the expansion.

Similarly, a rough estimate of INP/CN can be calculated by dividing the CPC measured concentration before the expansion by the maximum value of Welas2 measured concentrations of cloud particles larger than 30 µm. These values are provided in Table S1 for each expansion.

A pumped counterflow virtual impactor (PCVI) (Boulter et al., 2006) and an ice-selecting PCVI (IS-PCVI) (Hiranuma et al., 2016) were used to selectively sample cloud droplet and ice crystals, the residuals of which were counted with the CPC and

characterized by miniSPLAT and other instruments. PCVIs use a counterflow to reject smaller, interstitial particles, while particles that have enough inertia to pass through the counterflow are transmitted through the inlet. The ratio of the counterflow to the pumped flow changes the inlet cut-size. One advantage of using the PCVI system is to provide residual-laden flows, in which particles are virtually concentrated by a flow concentration factor (i.e., the ratio of input flow to output flow), to the downstream instruments. The IS-PCVI, which has a higher cut-size than other PCVIs (Hiranuma et al., 2016), is intended to

be used to reject both interstitial particles and small cloud droplets and allow the larger ice crystals through. The heated evaporation sections installed downstream of each PCVI are maintained at a temperature above 35 °C to evaporate water from the cloud particles leaving the residuals behind. The cut-sizes, enhancement factors, and transmission efficiencies for all of the expansions are given in Table S1. The cut-sizes are based on the measured number of particles transmitted through the PCVI

or IS-PCVI and the size distributions of cloud particles in the AIDA chamber. The transmission efficiencies for particles above the cut-size are 94% and 62% Expansions 1 and 2 respectively, respectively, (Gallavardin et al., 2008) and 91.5% for Expansion 3 (Hiranuma et al., 2016). The fraction of total cloud particles that were transmitted through the PCVI and IS-PCVI (Figure S1) was calculated using Eq. (1), where $F_{output}$ is the CVI output flow (lpm) and $F_{input}$ is the CVI input flow (lpm).

$$5 \quad \textit{Fraction of the droplets and ice crystals reached at the CVI}(t) = \frac{PCVI\ CPC(t) \times \frac{F_{output}}{F_{input}}}{Welas2(t)} \quad (1)$$

The fraction changes in time due to changing cut-sizes of the inlets and changing size distributions of the activated particles.

Vacuum aerodynamic diameters ($d_{va}$) and chemical compositions of individual aerosol particles and cloud residuals were measured with miniSPLAT, a single particle mass spectrometer, a detailed description of which is given elsewhere (Zelenyuk et al., 2015). Briefly, individual aerosol particles enter miniSPLAT through an aerodynamic lens inlet (Vaden et al., 2011a)

10  and are detected by light scattering in two optical stages 10.9 cm apart. The time it takes the particles to pass between the two continuous laser beams (532 nm Nd:YAG CrystaLaser, Model CL-300-LO) yields particle velocity, which for a given inlet pressure is related to particle $d_{va}$.

Given that the inlet pressure is changing during the expansions, polystyrene latex spheres (PSLs) of known sizes ranging from 73 nm to 993 nm were used to generate a pressure dependent size calibration curve over a range of pressures from 0.9 to 4

15  Torr. At normal atmospheric pressure the detection of spherical particles with diameters between 125 nm and 600 nm is very close to 100% (Zelenyuk et al., 2009;Zelenyuk et al., 2015). The detection efficiency of smaller particles decreases rapidly due to an increase in particle beam divergence and decrease in light scattering signal, resulting in the distinct sharp left edge of the $d_{va}$ distributions (Vaden et al., 2011b), with 50% cut off for particles with diameter of 85 nm (Zelenyuk et al., 2015;Vaden et al., 2011b). The detection of particles with $d_{va}$ larger than 600 nm decreases slowly because of the decrease in particles'

transmission efficiency through the aerodynamic lens inlet (Liu et al., 2007). Based on the prediction by computational fluid dynamics modeling, the transmission efficiency of larger particles is expected to further decrease at reduced inlet pressures (Liu et al., 2007). However, the results of experimental measurements on three different particle types do not show the predicted decrease in transmission efficiency for larger particles at lower pressure, indicating instead, an up to ~20% increase (Liu et al., 2007). The dual particle detection is also used to generate a trigger for the excimer laser (GAM Lasers Inc., Model

EX-5), operated at 193 nm with a constant laser energy of 1.0 ± 0.1 mJ/pulse, which ablates the particles and generates positive and negative ions. Single particle mass spectra are obtained with a dual-polarity, z-configuration reflectron time-of-flight mass spectrometer (Z-TOF, Tofwerk AG). miniSPLAT-measured particle number concentrations are calculated by dividing the particle detection rate at the first optical detection stage by the pressure-dependent sampling flow rate as described in detail in Vaden et al. (2011a). All data generated by miniSPLAT are processed, analyzed, and visualized using custom software

(Zelenyuk et al., 2008;Zelenyuk et al., 2006).

Suspensions of cultured Pseudomonas syringae and PF CGina 01 were prepared, as described in Hiranuma et al. (2016), by suspending $\sim10^9$ bacteria cells mL$^{-1}$ in sterile water and a small amount of Kings Base Agar. The solutions were atomized using a custom atomizer (Wex et al., 2015), dried by flowing through two diffusion driers (TOPAS, Model DDU 570), and injected into the AIDA chamber prior to expansions.

## 3 Results and Discussion

### 3.1 Pseudomonas Syringae Bacteria Characterization

The AIDA cloud chamber was preconditioned overnight to reach the desired experimental temperature. Before the expansions a suspension of bacteria (Pseudomonas syringae for expansion 1 and PF CGina for expansions 2 and 3) was aerosolized, dried, and the resulting particles were introduced into the chamber. Figure 1 shows the calculated volume equivalent diameter ($d_{ve}$) size distributions of particles present in AIDA before expansion 1, which was obtained by combining measurements by the SMPS (black line) and the APS (red line). The size distribution is bimodal with a smaller size mode that peaks at ~70 nm and a broad peak at ~800 nm. The miniSPLAT-measured vacuum aerodynamic size ($d_{va}$) distribution, shown in Figure 1b, is also bimodal with modes at 180 nm and ~850 nm. The difference between the small particle peak positions of the measured $d_{va}$ and $d_{ve}$ size distributions is due to particle density ($\rho > 1$ g cm$^{-3}$) and the miniSPLAT's small particle detection limit (50% cut off at 85 nm, marked with a green line in Figure 1a) (Vaden et al., 2011b). This type of bimodal size distribution has been observed previously with other aerosolized suspensions of cultivated bacteria (Wex et al., 2015;Wolf et al., 2015;Möhler et al., 2008). The large size mode corresponds to intact/whole bacteria cells and the smaller mode is composed of bacterial fragments mixed with agar.

Representative positive and negative ion mass spectra (MS) of the two particle modes are shown in Figure 2. The reference MS of the small particle mode includes MS of particles smaller than 300 nm and that of the large particle mode/intact bacteria includes the MS of particles larger than 700 nm. Comparison between the reference MS of the two particle modes shows that nearly all the peaks are present in the MS of both modes. However, the positive ion MS of the two modes exhibit significantly different relative peak intensities, while the negative ion MS are very similar. The positive ion MS indicate the presence of many organics and metal ions, including carbon containing fragments ($^{12}C^+$, $^{15}CH_3^+$, $^{28}CO^+$), $^{41}CH_3CO^+$, $^{44}CO_2/CH_3COH^+$), ammonium ($^{18}NH_4^+$), organic nitrogen ($^{28}HCNH^+$, $^{30}H_2NCH_2/NO^+$, $^{70}HNCH(CH_2)_3^+$, $^{85}C_5NH_{10}^+$), sodium ($^{23}Na^+$), potassium ($^{39/41}K^+$), calcium ($^{40}Ca^+$), and calcium oxides ($^{56}CaO^+$, $^{113}(CaO)_2H^+$). The negative ion MS show the presence of organic nitrogen ($^{26}CN^-$, $^{42}CNO^-$), chloride ($^{35/37}Cl^-$), organics ($^{45}HCO_2^-$, $^{50}C_3N^-$, $^{71}C_3H_3O_2^-$), and phosphates ($^{63}PO_2^-$, $^{79}PO_3^-$, $^{97}H_2PO_4^-$). The ions were identified via comparison to previous studies on bacteria (Pratt et al., 2009;Zawadowicz et al., 2017;Wolf et al., 2015;Fergenson et al., 2004;Srivastava et al., 2005;Czerwieniec et al., 2005). The positive ion MS for intact cells show significantly higher intensities for the organic peaks ($^{12}C^+$, $^{15}CH_3^+$, $^{28}CO/HCNH^+$, $^{30}H_2NCH_2/NO^+$, $^{44}CO_2/CH_3COH^+$, $^{70}HNCH(CH_2)_3^+$, $^{85}C_5NH_{10}^+$) as compared to those of the small particle mode, which are dominated by the potassium peak. The

same MS are presented in Figure 3 in expanded scale together with the MS of pure agar particles. Comparison between MS of agar and the small particle mode shows their positive ion MS have the same peaks just with varying ion intensities and the absence of peak at m/z 131 (possibly, $C_7H_{17}NO$) in the agar. The agar negative ion MS have sulfate peaks ($^{64}SO_2^-$), $^{81}HSO_3^-$, $^{97}HSO_4^-$) and lack some organic ions ($^{45}HCO_2^-$, $^{71}C_3H_3O_2^-$) as compared to the bacteria MS. The size distributions and MS

provide direct evidence that the smaller particles are composed of bacterial fragments mixed with agar, while the larger size particle mode corresponds to intact bacteria cells. The MS of intact bacteria shown in Figures 2 and 3 are similar to previously published single particle mass spectra of laboratory-generated Pseudomonas syringae acquired by two other single particle mass spectrometers (Pratt et al., 2009;Zawadowicz et al., 2017). The negative ion MS presented here and in previous studies are virtually the same, while the positive ion MS presented in this study exhibit significantly higher intensity in organic peaks,

most likely due to the differences in the ablation laser power or wavelength (266 nm in Pratt et al. (2009) and 193 nm in this study and Zawadowicz et al. (2017)). Moreover, these studies do not mention the presence of two particle size modes nor distinguish between their mass spectra.

The size distributions shown in Figure 1a indicate a total number concentration of particles in the chamber of ~2,500 particles cm$^{-3}$and ~100 particles cm$^{-3}$, or 4%, are intact bacteria. In comparison, the $d_{va}$ size distribution (Figure 1b) yields 7% for the

fraction of intact bacteria, since more than half of the particles in the smaller size mode are too small to be detected by miniSPLAT. It is important to point out that the size, number concentration, and the relative fraction of the small particles containing bacterial fragments can vary from experiment to experiment and are determined by the solution/suspension concentrations of agar, intact bacteria, and cell fragments, as well as how the bacteria culture is grown and prepared (Wolf et al., 2015), and may change with the age of the solution/suspension.

**3.2 Cloud Formation and Cloud Droplet Characterization**

Cloud formation was induced by lowering the pressure in the AIDA chamber, using active pumping, which lowered the temperature and increased the RH, shown in Figure 4a. As the chamber became supersaturated with respect to both ice and water, cloud droplets and ice crystals formed, which is illustrated in Figure 3b with a false color plot of the welas2-measured size distribution as a function of expansion time. The number concentration of cloud droplets and ice crystals (particles larger

than 5 µm, plotted as a black line) quickly reached ~900 particle cm$^{-3}$ and then decreased slowly to 400 particles cm$^{-3}$. An estimate of the fraction of particles that activated as cloud droplets (CCN/CN) yields 0.45, indicating that less than half of the particles were incorporated into droplets. After ~500 sec, as pumping stopped, the temperature increased, RH dropped, and the droplet number concentration decreased mainly due to water evaporation. During the expansion, the vast majority of the droplets remained smaller than 14 µm. The few ice crystals that formed are apparent in the figure as a second mode of much

larger particles (marked in Figure 4b) mainly visible at ~100 to 200 sec, after which point their numbers rapidly decreased mainly because of settling losses.

Ice crystal concentrations derived from welas2-measured number concentrations of particles larger than 30 µm (pink line in Figure 4c) reach a maximum of 3 particles cm$^{-3}$, or more than 200 times lower than droplets. An estimate of the fraction of particles that were incorporated into ice crystals shows that INP/CN ≈ 0.0014. Figures 4b and 4c show that during Expansion 1 the PCVI transmitted predominantly droplets, the residuals of which were characterized by the CPC and miniSPLAT.

The $d_{va}$ size distribution of cloud residuals (Figure 5) is virtually the same as the size distribution of particles in the chamber before the expansion except that the intact bacteria, i.e. the large particle mode, is "missing" from the size distribution of the cloud residuals.

In addition to the $d_{va}$ size distribution, miniSPLAT measured positive and negative ion MS of individual cloud residuals, which are presented in Figure 6 superimposed on the reference MS of intact bacteria (large mode particles) (Figure 6a and b) and on
the reference MS of the small particle mode composed of agar and bacterial fragments (Figure 6c and d) measured before the expansion. Comparison between the three particle types shows that the intact bacteria have higher relative intensities of many organic and nitrogen-containing ($^{15}CH_3^+$, $^{18}NH_4^+$, $^{28}HCNH^+$, $^{30}H_2NCH_2/NO^+$, $^{44}CO_2^+$, $^{70}HNCH(CH_2)_3^+$, $^{85}C_5NH_{10}^+$) and calcium ions ($^{40}Ca^+$, $^{56}CaO^+$) than the cloud residuals, but the small particle mode has nearly the same MS as the cloud residuals.. The small differences between the MS of residuals and the small particle mode suggest a slightly higher content of organics in the
residuals, indicated by higher relative intensities of $^{28}HCNH^+$, $^{30}H_2NCH_2/NO^+$, and $^{44}CO_2^+$, which is due to the fact that the MS of residuals includes all detected particles, while the reference MS of the small particle mode includes only particles smaller than 300 nm. These MS differences will be further examined below in a detailed analysis of data obtained during Expansion 2, where the observed differences are larger and, therefore, easier to analyze. Taken together with the size distributions, these data provide direct evidence that under these experimental conditions, from the two particle types present
in the chamber, particles composed of bacterial fragments mixed with agar preferentially activated to form cloud droplets.

**3.3 PF CGina Bacteria and Cloud Droplet Characterization**

Expansion 2 was performed in the same manner as Expansion 1, except that different bacteria, PF CGina, was used. Figure 7a presents the size distributions of particles in the chamber prior to Expansion 2, showing two particle modes, the small size
mode, composed of agar and bacterial fragments, which peaks at $d_{ve}$≈60 nm, and the intact bacteria, which peaks at $d_{ve}$≈700 nm. There were 1,800 particles cm$^{-3}$, ~4% of which were intact bacteria. Similarly, the $d_{va}$ size distribution, shown in Figure 7b shows the small particle mode peaking at $d_{va}$≈160 nm, composed of agar and bacterial fragments, and the intact bacteria mode, which peaks at $d_{va}$≈850 nm and represents ~9% of particles detected and sized prior to Expansion 2. Comparison between the reference MS of the two particle modes present before Expansion 2 is presented in Figure 8, where the small mode
includes particles smaller than 300 nm, and the large mode represents particles larger than 700 nm. As in the case of Expansion 1, the MS show that the small particles are composed of agar and bacterial fragments. The difference between the two positive

ion reference MS for PF CGina are very similar to the pattern observed for Pseudomonas syringae bacteria (Figure 2) and contain all of the same ions.

The temporal evolution of pressure, temperature, and RH during Expansion 2 are shown in Figure 9a. The welas2-measured size distributions of cloud particles (Figure 9b) reveal that most of the activated particles were liquid droplets smaller than 10 µm, and only a small number of cloud particles were ice crystals. The average droplet and ice crystal concentrations (Figure 9c) were ~730 particles cm$^{-3}$ and ~1 particle cm$^{-3}$, respectively, with ice crystal peak concentrations reaching ~3 particles cm$^{-3}$ at 150 to 250 seconds from the start of the expansion. As a result, particles transmitted through the PCVI to the CPC and miniSPLAT during Expansion 2 were predominantly liquid droplets. The CPC- and miniSPLAT-measured number concentrations of particles transmitted by the PCVI, Figure 9c, show rapid increases early in the expansion and slower decreases thereafter, displaying the same behavior as in Expansion 1. A lower percentage of particles were activated into droplets (30 %) than in the first expansion, but roughly the same percentage were activated into ice crystals (0.16%).

The miniSPLAT-measured $d_{va}$ distributions of cloud residuals (blue) and particles in the chamber prior to the expansion (green) are shown in Figure 10. As with Expansion 1, the $d_{va}$ distribution of cloud residuals is nearly identical to that of the small particle mode present in the chamber before the expansion. An expanded scale of the intact bacteria size region, shown in the figure inset, indicates no detectable intact bacteria in cloud residuals.

The average positive ion MS of cloud residuals is shown in Figure 11. For comparison, it is superimposed on the reference MS of the small particle mode (Figure 11a) and intact bacteria (Figure 11b). The MS of residuals shows that they contain slightly more organics ($^{12}C^+$, $^{15}CH_3^+$, $^{28}HCNH^+$, $^{30}H_2NCH_2/NO^+$, $^{44}CO_2^+$, $^{70}HNCH(CH_2)_3^+$, $^{85}C_5NH_{10}^+$) than the particles that were included in the reference MS of the small particle mode. It is important, however, to note that the residual MS includes all particles, while the reference MS contains only particles smaller than 300 nm. A possible explanation for the observed difference between the two MS is that the organic content of the small particle mode increases with particle size, i.e. larger particles contain more or larger bacterial fragments and relatively less agar. Indeed, Figure 12 shows that the fraction of characteristic organic peaks (m/z 28 and 44) relative to K$^+$, which serves as a simple qualitative measure of the relative fraction of bacterial fragments and agar in these particles, is increasing with particle size. The figure includes the $d_{va}$ distribution of cloud residuals (via a dashed line) for reference. For comparison, the same analysis shows that the K$^+$ peak contributes 86% of the mass spectral intensity for pure agar and 13% for intact bacteria. To examine this further, we compare in Figure S2a the MS of cloud residuals smaller than 300 nm to the reference MS of the small particle mode measured before the expansion and find them to be nearly identical. This indicates that they are bacterial fragments mixed with agar. The MS of cloud residuals that are larger than 300 nm have greater ion intensities for the organic peaks ($^{28}CO/HNCH^+$, $^{44}CO_2/CH_3COH^+$, $^{70}HNCH(CH_2)_3^+$, $^{85}C_5NH_{10}^+$) than the bacterial fragments mixed with agar, but slightly smaller ion intensities than those of the reference MS of intact bacteria, as shown in Figure S2b. These differences in ion intensities are not sufficient to eliminate the possibility that some of the larger cloud residuals could be smaller intact bacteria.

Overall, the $d_{va}$ size distributions and MS of cloud residuals are like those of the small particle mode, although the larger residuals, which contain more organics, could be smaller intact bacteria. Nevertheless, the data clearly show that the intact bacteria contribute a significantly lower fraction, if at all, to the cloud residuals than they contribute to the total particle population present before the expansion. In all respects, the miniSPLAT data for PF CGina01 are consistent with those for Pseudomonas syringae (Expansion 1) and confirm the preferential droplet activation of particles composed of bacterial fragments mixed with agar compared to intact bacteria.

## 3.4 PF CGina Ice Crystal and Droplet Characterization

Expansion 3 followed the same pumping strategy and expansion temperatures as the previous expansions, but was conducted on the PF CGina particles which remained in the AIDA chamber after Expansion 2. The $d_{ve}$ size distributions of these particles, shown in Figure 7a as dashed lines, are nearly identical to those measured before Expansion 2, despite a decrease in the total particle number concentration. During Expansion 3, the cloud residual particles, sampled by the CPC and miniSPLAT, were transmitted through a different inlet, the IS-PCVI, in order to reduce the number of droplets transmitted to allow for characterization of ice crystal residuals.

Changes in chamber pressure, temperature, and RH for Expansion 3 are shown in Figure 13a. A false color plot of the activated particles size distribution measured by welas2, as a function of time, is presented in Figure 13b. Figure 13c displays the welas2-measured total number of activated particles as a function of time (black), which reaches a maximum of ~700 particles cm$^{-3}$. Before the expansion, 1456 cm$^{-3}$ particles were present in the chamber indicating that 47% of the particles activated as CCN. In addition, it shows the number concentrations of cloud particles with optical diameters larger than 15 µm but smaller than 20 µm (light blue) and the number of particles larger than 20 µm (pink). The former may correspond to the number of droplets transmitted by the IS-PCVI, while the latter may be relevant to transmitted ice crystals. Comparison between the two shows that the IS-PCVI transmitted nearly equal numbers of ice crystals and droplets to miniSPLAT and the CPC. A small percentage of particles (0.04%) formed ice crystals. miniSPLAT and the CPC-measured particle number concentrations are also shown in Figure 13c. Both instruments show a rapid rise in the concentration of cloud residuals, followed by a significant drop in concentration with time. Early in the expansion, most of the transmitted particles are droplets, but later, between 302 and 557 seconds after the expansion started, the number of droplets and ice crystal residuals becomes comparable, as shown in Figure 13c.

Two particle modes are present in the $d_{va}$ size distribution of particles in the chamber before Expansion 3 (Figure 14, green trace). Figure 14 also shows the $d_{va}$ size distributions of all cloud residuals detected throughout the expansion (blue) and cloud residuals detected late in the expansion (302-557 seconds, light blue), when the number concentration of droplets and ice crystals were approximately equal. Both cloud residual size distributions are nearly the same as that of the small particle mode and show no distinct peak in the size region of intact bacteria.

The average MS of particles sampled through the IS-PCVI throughout the expansion and later in the expansion are shown in Figure 15. The fact that the two MS are virtually identical suggests that the compositions of particles that remain as droplets and those that activate into ice crystals are not different, which is consistent with the finding that the size distributions are the same. The residual size distributions and compositions combined again confirm that the majority of particles that served as CCN and INPs were bacterial fragments mixed with agar. While there is no clear indication that intact bacterial cells were activated, it cannot be ruled out that larger cloud residuals were small intact bacteria mixed with agar.

These results are consistent with previous studies that show that whole cells are not necessary for ice formation because IN active proteins and protein complexes can serve as INPs (Lindow et al., 1989;Maki and Willoughby, 1978;Hartmann et al., 2013;Govindarajan and Lindow, 1988). However, it has generally been assumed that whole cells will also nucleate ice (Möhler et al., 2008;Wex et al., 2015;Lindow et al., 1989) and are more active than a single IN active protein (Govindarajan and Lindow, 1988) or bacterial fragment (Yankofsky et al., 1981;Hartmann et al., 2013). The explanation for enhanced IN activity of intact bacteria is that IN active proteins collect in the cell membrane (Lindow et al., 1989;Govindarajan and Lindow, 1988) and become concentrated on the surface of intact cells thus enhancing their IN activity. However, the data presented here show that bacterial fragments mixed with agar preferentially activate as droplets and are the only particles observed in ice residuals under these experimental conditions.

## 4 Conclusions and Implications for Future Studies

This paper presents the results of measurements of particle number concentrations, size distributions, and compositions of aerosol particles produced from two types of bacteria particles in the AIDA chamber for three separate expansions, before and after the expansions. In all cases, the data show the presence of two distinct particle types in the chamber: particles larger than ~ 700 nm that are intact bacteria, and small particles, with a $d_{ve}$ distribution that peaks at ~65 nm, that are composed of bacterial fragments mixed with agar. The MS show that the small and large particles have distinct MS, representing differences in particle compositions. The small particles are composed of a mixture of bacterial fragments and agar, as determined by comparison with the MS of pure agar particles and intact bacteria. In addition, a detailed analysis of the MS of all particles shows that the organic content of particles relative to potassium increases with particle size, indicating that smaller particles have a larger fraction of agar, whereas larger particles have more organics, i.e. bacterial fragments.

The concentration of small particles in the chamber was about thirty times higher than that of intact bacteria for all expansions. The two particle modes in the chamber were detected before and after three expansions, providing direct evidence that intact bacteria remain intact throughout the expansion and do not burst upon freezing or drying.

In the three expansions presented here, cloud activation occurred between –5 and -12 °C to form cloud droplets, a small fraction of which froze into ice crystals (0.0004-0.0016). Cloud droplet concentrations were 30-47% of the total particle number concentration in the AIDA chamber, which is significantly higher than the concentration of intact bacteria (4%).

The cloud particles were separated with the PCVI or IS-PCVI, and their residuals were characterized with miniSPLAT. The data in all three expansions present virtually the same picture. The $d_{va}$ size distributions of cloud residuals were devoid of the large particle/ intact bacteria mode and were nearly the same as the miniSPLAT-measured $d_{va}$ distributions of the small particle/ bacterial fragment mode acquired before and after the expansions, providing direct evidence that bacterial fragments mixed with agar activated to form either droplets or ice crystals were with much higher efficiency than intact bacteria.

MS analysis of the cloud residuals confirms that, while both intact bacterial cells and bacterial fragments mixed with agar were present in the AIDA chamber before and after the expansions, the latter represented a much larger fraction of particles that served as CCN and as INPs than present in the chamber, which is consistent with the $d_{va}$ size distributions of cloud residuals. These results were replicated in three expansions, for two different bacteria strains, Pseudomonas syringae and PF CGina 01, suggesting that this behavior may be representative of bacteria in general, or at least of cultivated bacteria used in most laboratory experiments. This is the first study to characterize the size distributions and mass spectra of bacteria cloud residuals sampled from the AIDA chamber. The detailed size distributions downstream of the PCVI and IS-PCVI enabled identification of the particle types in the AIDA chamber that were activated into cloud droplet and ice crystals.

It is important to note that at the activation temperatures and RH$_w$ of the three expansions presented here, particles first activate as CCN to form cloud droplets and ice crystals form by immersion freezing. If the reports suggesting that some strains of intact bacteria are weakly hygroscopic (Franc and DeMott, 1998;Sharma and Rao, 2002) are correct, it is not surprising to find that intact bacteria have limited CCN activity, and by necessity, a negligible amount can serve as INP in the immersion mode. Drop freezing and other bulk immersion freezing techniques bypass the issue of limited hygroscopicity by submerging whole cells in a droplet or well of water, artificially overcoming the barrier of CCN activation. These distinctions are important to consider when relating laboratory INP measurements to estimations of ice formation in atmospheric clouds. A recent modeling study has shown that competition for water vapor can inhibit ice formation in clouds due to the most CCN-active particles taking up all the water and leaving the more IN active, but less CCN active, particles as interstitial aerosol and thus, unable to form ice in clouds (Simpson et al., 2017). The study presented here shows that, in contrast to intact bacteria, particles composed of bacterial fragments mixed with agar are CCN active and a fraction of their resulting cloud droplets freeze to make ice crystals. There was possibly competition for water vapor in these expansions as only ~30-47% of particles formed droplets. Therefore, the most hygroscopic particles would be the ones to activate as CCN.

If agar aids in transforming bacterial fragments into effective CCN and INP by enhancing their hygroscopicity, then it is important to quantify the role agar plays in CCN and IN activity of bacteria by conducting experiments on bacterial samples that do not contain agar. Pure agar particles do not freeze to form ice crystals at the temperature range examined in this study

(Hiranuma and Möhler, 2018), indicating that the bacterial fragments served as INPs in these expansions even if the agar enhanced their hygroscopicity. Most laboratory studies on bacteria use cultivated bacteria to make measurements that are then extrapolated to atmospheric conditions. However, these measurements may not accurately reflect the CCN or IN activity of non-cultured bacteria under atmospheric conditions.

*Data availability.* The data used in this manuscript are available at https://dtn2.pnl.gov/data/release/2018_Suski_et_al_Bacteria_Paper/.

*Competing interests.* The authors have no conflict of interest.

*Special issue statement.* This article is part of the special issue "Fifth International Workshop on Ice Nucleation (FIN)". It is not associated with a conference.

*Acknowledgments.* Support for AZ, DMB, and KS was provided by the U.S. Department of Energy (DOE) Office of Biological and Environmental Research (OBER) Atmospheric Research Systems Program (ASR). Development of miniSPLAT was funded by the DOE Office of Science, Office of Basic Energy Sciences, Division of Chemical Sciences, Geosciences & Biosciences and EMSL User Facility sponsored by the DOE OBER and located at Pacific Northwest National Laboratory. The valuable contributions of the FIN organizers, their institutions, and the FIN-1 Workshop science team are also gratefully

acknowledged. OM and NH thank the Engineering and Infrastructure group members at KIT IMK-AAF (Georg Scheurig, Tomasz Chudy, Rainer Buschbacher, Olga Dombrowski, Steffen Vogt, Jens Nadolny and Frank Schwarz) for their technical support during FIN-1. The AIDA work was partly funded by the Helmholtz Association through its research programme 'Atmosphere and Climate (ATMO)'.

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

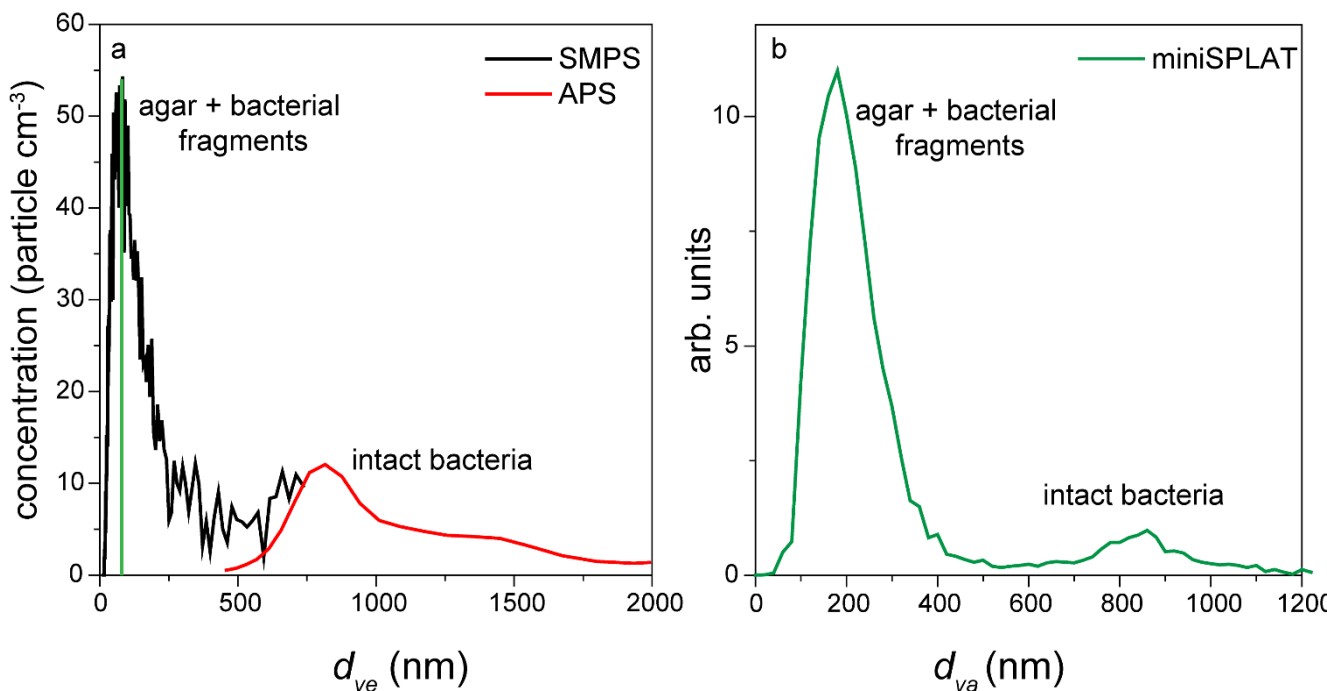

**Figure 1: Size distributions of an aerosolized suspension of Pseudomonas syringae in the AIDA cloud chamber before Expansion 1, calculated based on the SMPS and the APS measurements (a) and miniSPLAT (b). The miniSPLAT small particle detection limit is denoted by the green line in (a).**

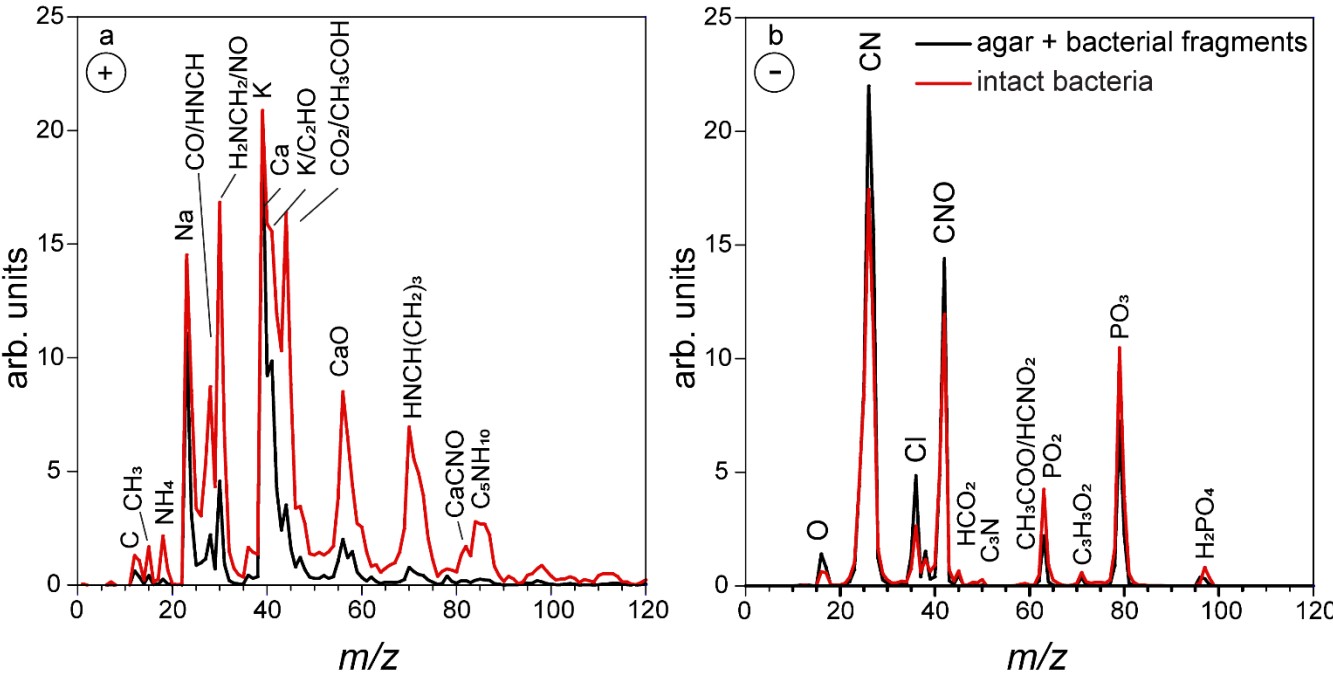

**Figure 2: Positive (a) and negative (b) average mass spectra of the particles in the AIDA chamber, measured before Expansion 1. Reference mass spectra of the small size mode, made up of bacterial fragments mixed with agar, which includes particles smaller than 300 nm, are marked in black, and the mass spectra of the large size mode particles ($d_{va}$ > 700 nm), composed of intact Pseudomonas syringae bacterial cells, are marked in red.**

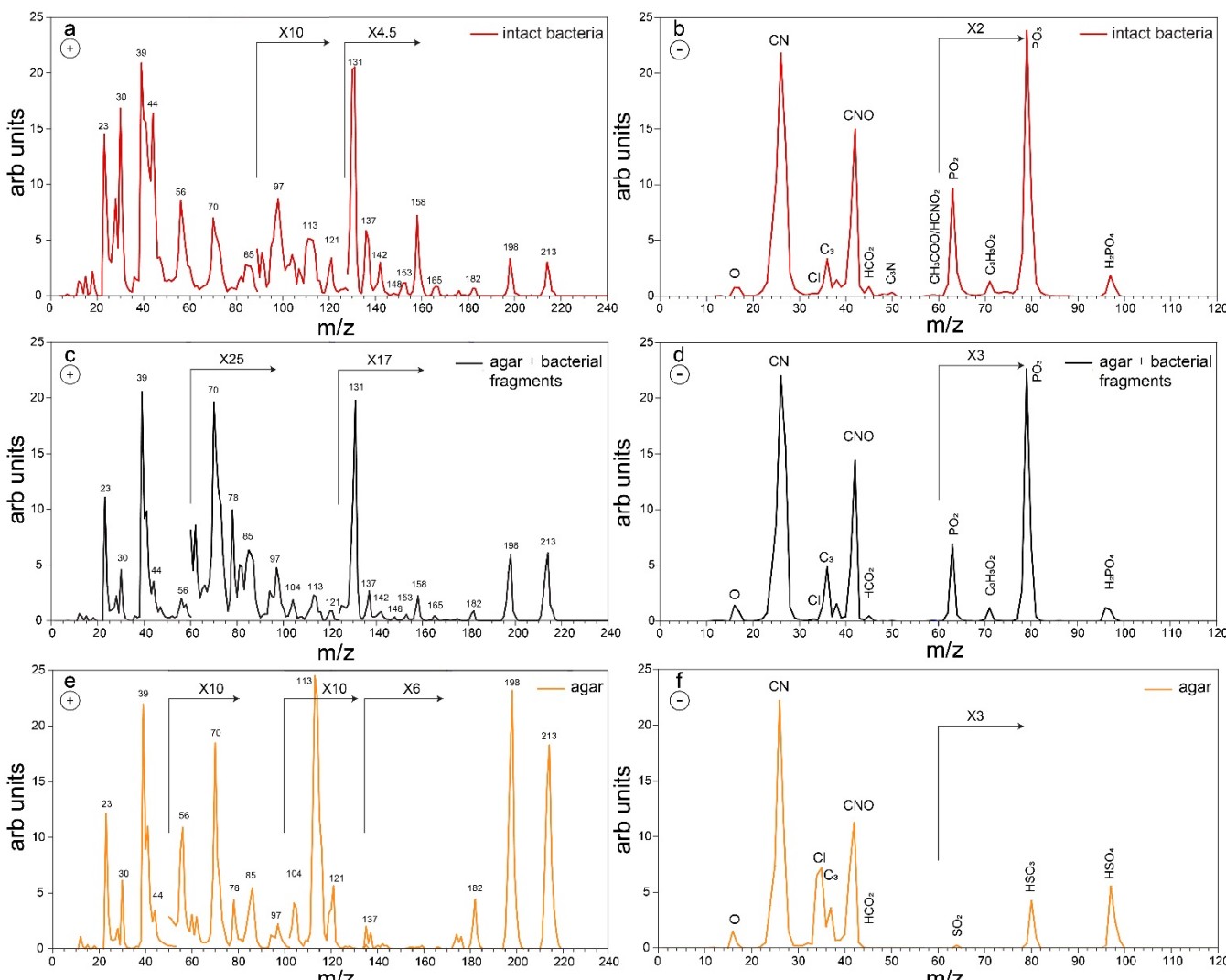

**Figure 3: Positive (a) and negative (b) average mass spectra (MS) of intact Pseudomonas syringae bacteria measured before Expansion 1; Positive (c) and negative (d) average MS of the small particles mode, measured before Expansion 1; Positive (e) and negative (f) average MS of pure agar particles measured in a separate experiment.**

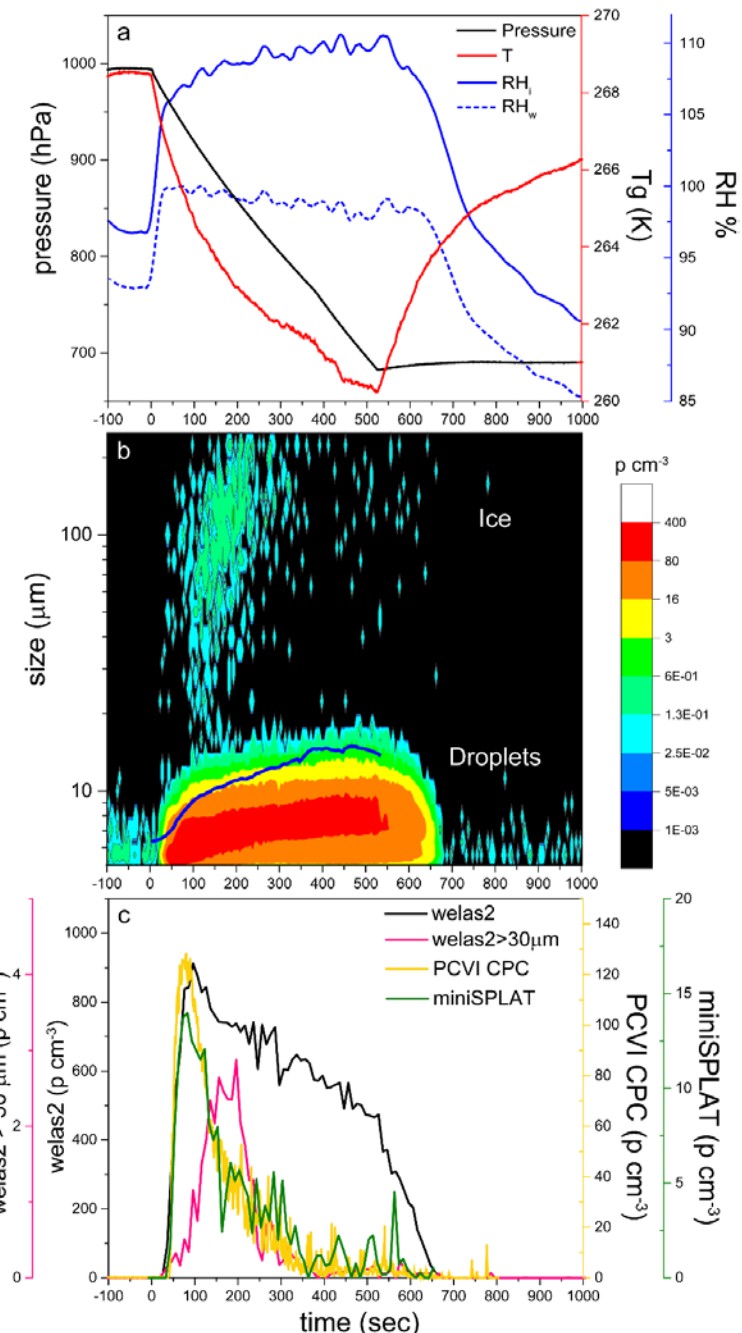

**Figure 4: Data from Expansion 1 with Pseudomonas syringae bacteria. (a) The pressure (black) and the temperature of the gas (red) inside the AIDA chamber. The measured RH with respect to ice (RHi) and water (RHw) are indicated in solid and dashed blue lines, respectively; (b) the size distributions of the particles measured with welas2 during the expansion. Particles smaller than ~15 μm are liquid droplets, while particles with larger optical diameters are ice. The PCVI cut-size is shown by the blue trace; (c) The welas2-measured total particle number concentrations (black) and the number concentrations of particles larger than 30 μm (pink). The number concentrations of particles transmitted by the PCVI and measured by CPC and miniSPLAT are marked in yellow and green, respectively.**

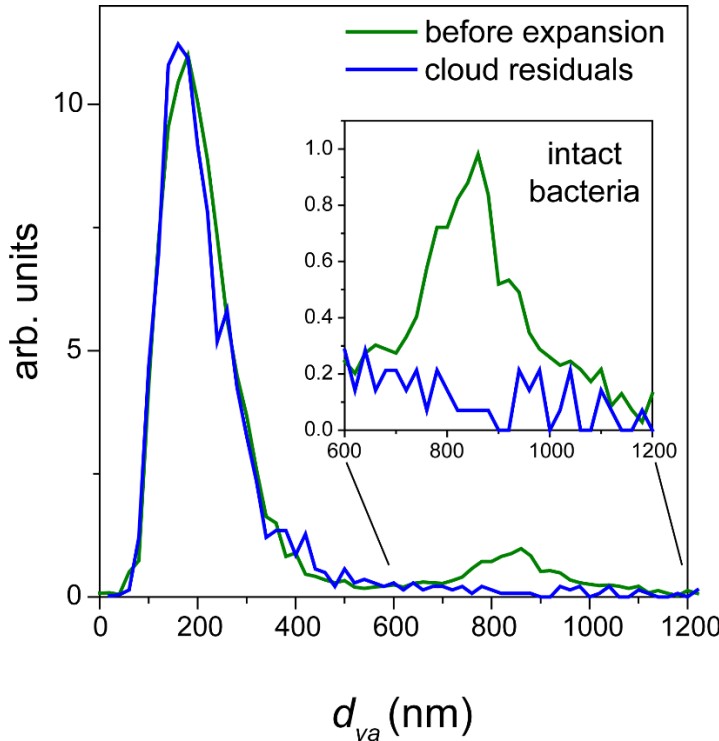

**Figure 5: Normalized miniSPLAT-measured** $d_{va}$ **size distributions of Pseudomonas syringae particles in the AIDA chamber before Expansion 1 (green) and of cloud residuals transmitted through the PCVI during the expansion (blue). The inset presents in expanded scale the size distributions of the intact bacteria, showing that cloud residuals do not exhibit the distinct peak for intact bacteria.**

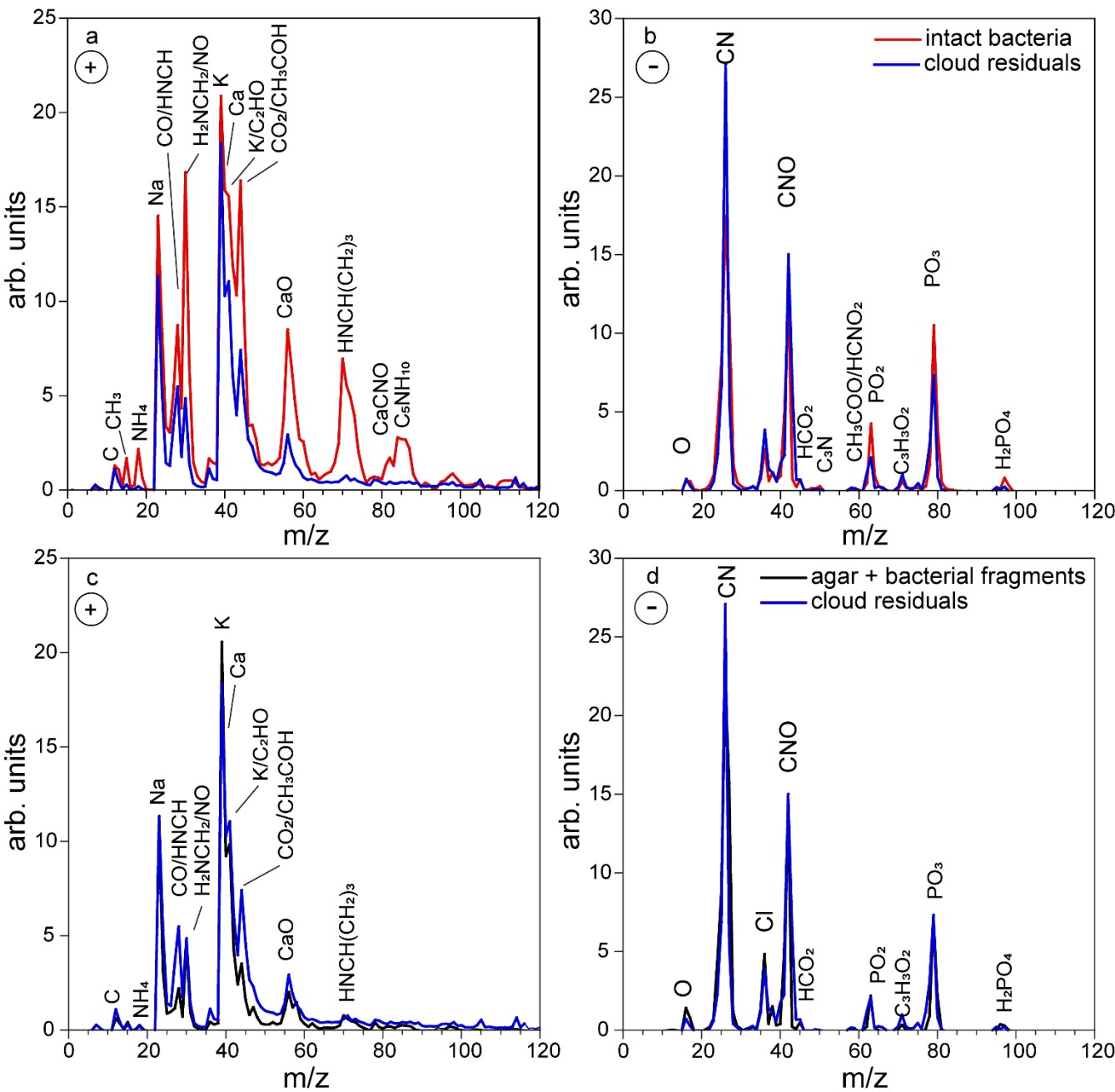

**Figure 6: Positive (a) and negative (b) ion average mass spectra of cloud residuals sampled during Expansion 1 (blue) superimposed on the average MS of intact bacteria (red); (c) and (d) The same cloud residuals average MS superimposed on the reference average MS of the small particle mode, composed of Pseudomonas syringae bacterial fragments mixed with agar (black).**

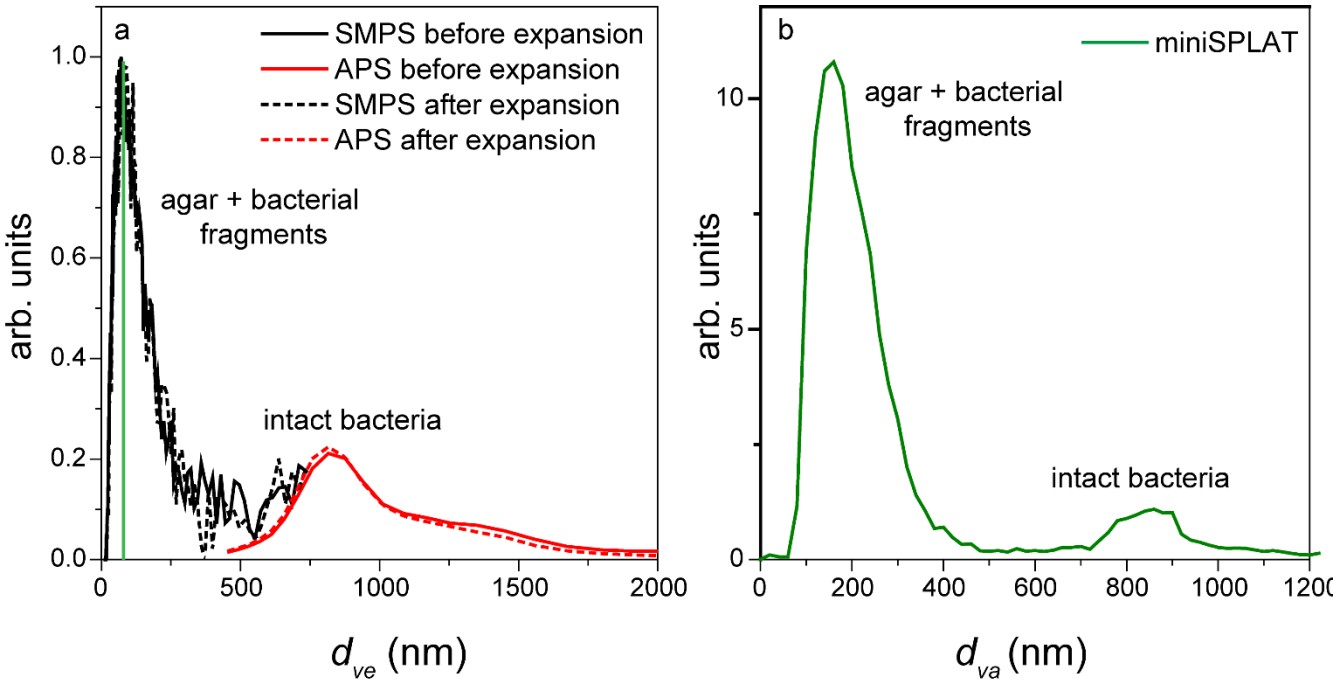

**Figure 7: (a) Size distributions of an aerosolized suspension of PF CGina particles in the AIDA cloud chamber before and after Expansion 2 calculated from the SMPS and the APS measurements; (b) miniSPLAT-measured size distribution of PF CGina particles in the chamber before Expansion 2. The miniSPLAT small particle detection limit is denoted by the green line in (a).**

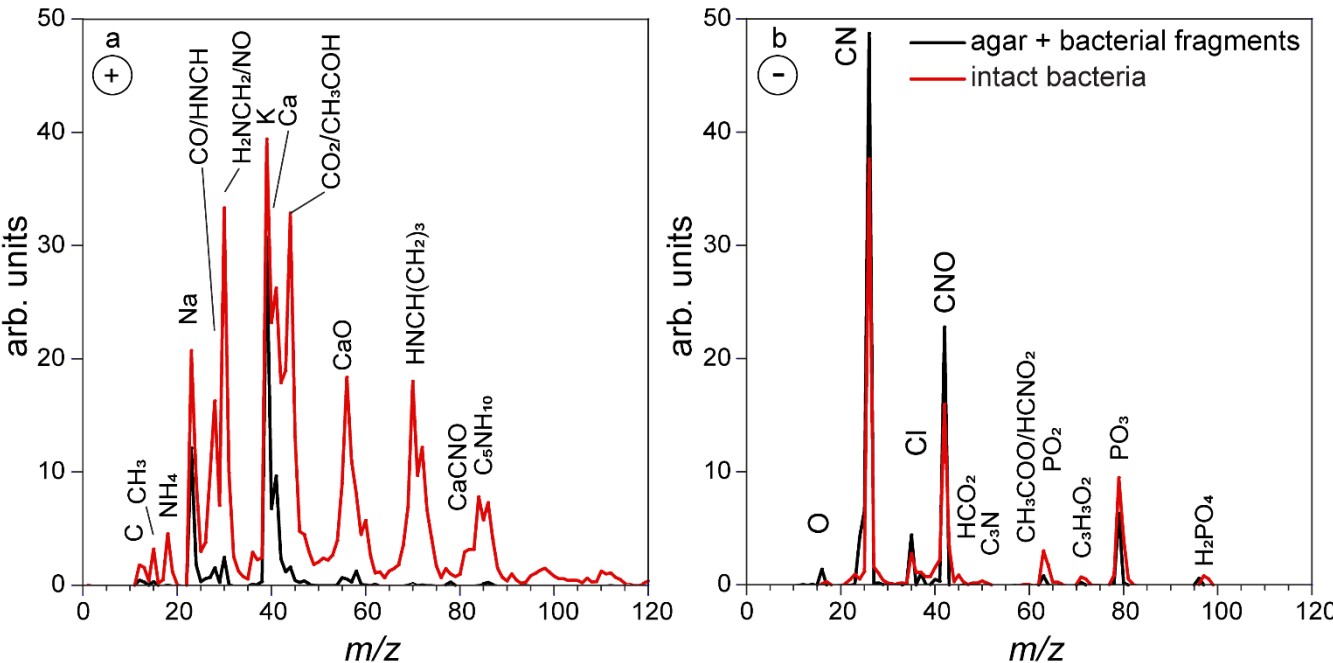

**Figure 8: Positive (a) and negative (b) average mass spectra of the particles in the AIDA chamber, measured before Expansion 2. Reference mass spectra of the small size mode, made up of particles smaller than 300 nm (black), and the mass spectra of the large size mode (>700 nm) composed of intact PF CGina bacterial cells are marked in red.**

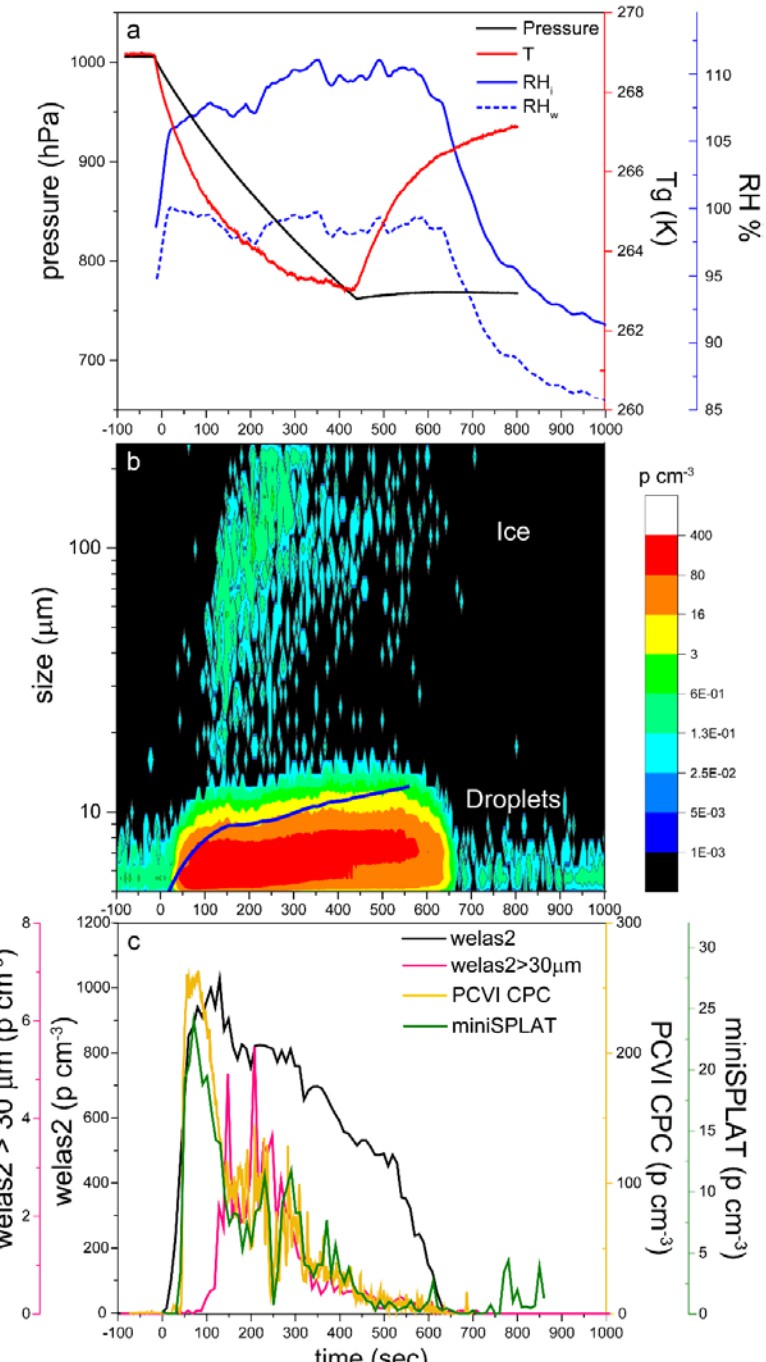

**Figure 9: Data from Expansion 2 with PF CGina bacteria. (a) The pressure (black) and the temperature of the gas (red) inside the AIDA chamber. The measured RH with respect to ice (RHi) and water (RHw) are indicated in solid and dashed blue lines, respectively; (b) the size distributions of the particles measured with welas2 during the expansion. Particles smaller than ~15 μm are liquid droplets, while larger particles are ice. The PCVI cut-size is shown by the blue trace; (c) The welas2-measured total particle number concentrations (black) and the number concentrations of particles larger than 30 μm (pink). The number concentrations of particles transmitted by the PCVI and measured by CPC and miniSPLAT are marked in yellow and green, respectively.**

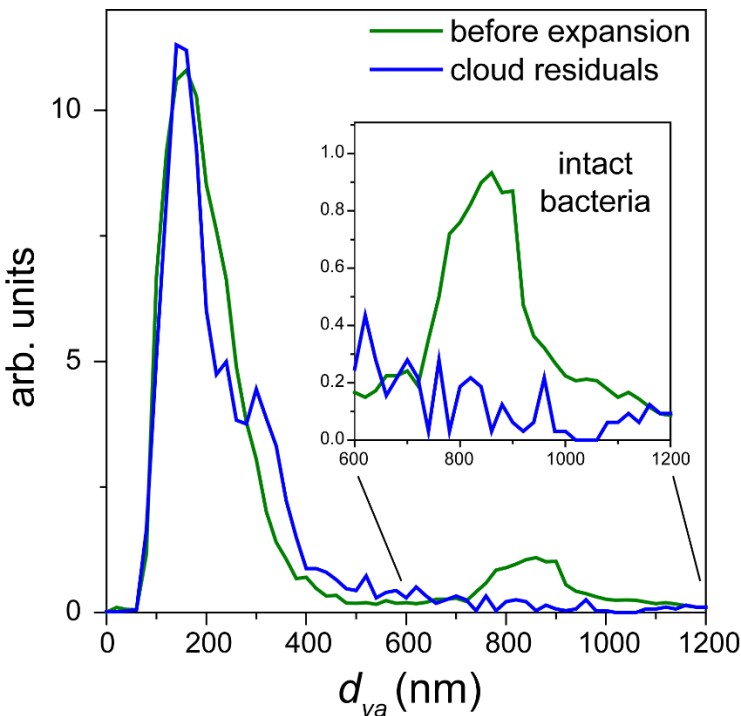

**Figure 10: Normalized miniSPLAT-measured** $d_{va}$ **size distributions of PF CGina particles in the AIDA chamber before Expansion 2 (green) and of cloud residuals transmitted through the PCVI during the expansion (blue). The inset presents in expanded scale the size distributions of the intact bacteria, showing that nearly no distinct peak for intact bacteria.**

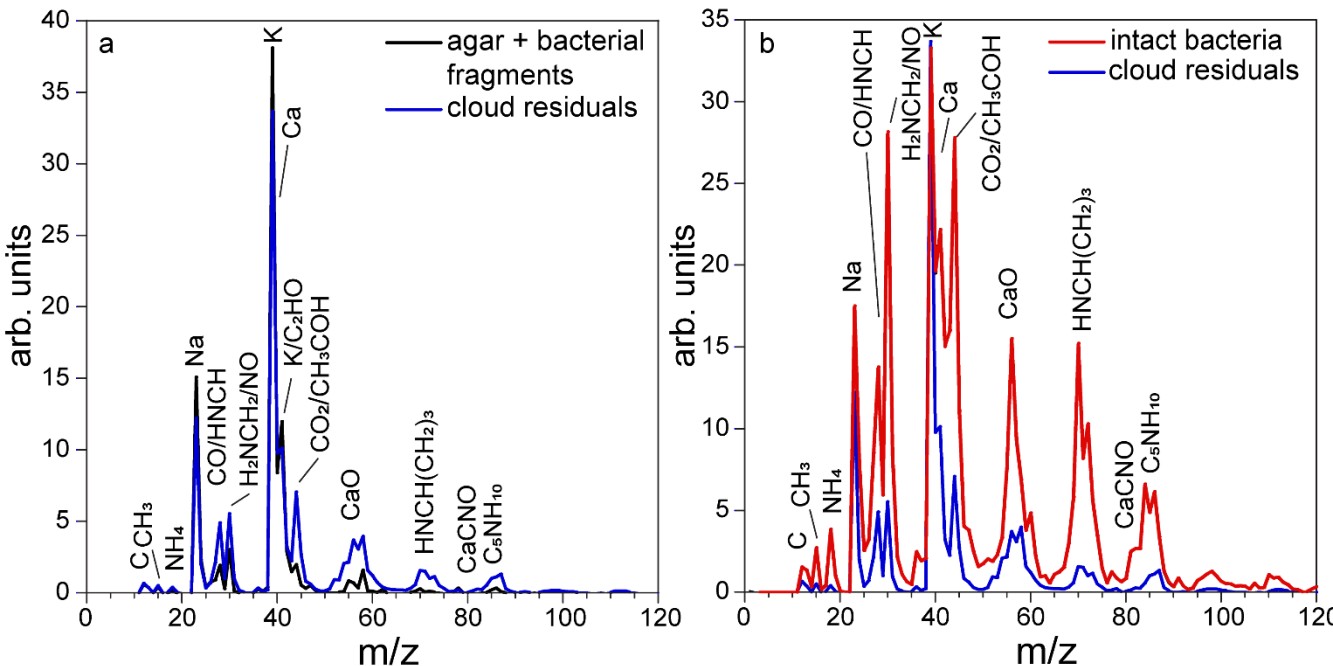

**Figure 11:** (a) Average reference MS of the small particle mode, composed of agar and PF CGina bacterial fragments, (black) and the average MS of cloud residuals acquired during Expansion 2 (blue); (b) average MS of intact bacteria (red) and the average MS of cloud residuals acquired during Expansion 2 (blue).

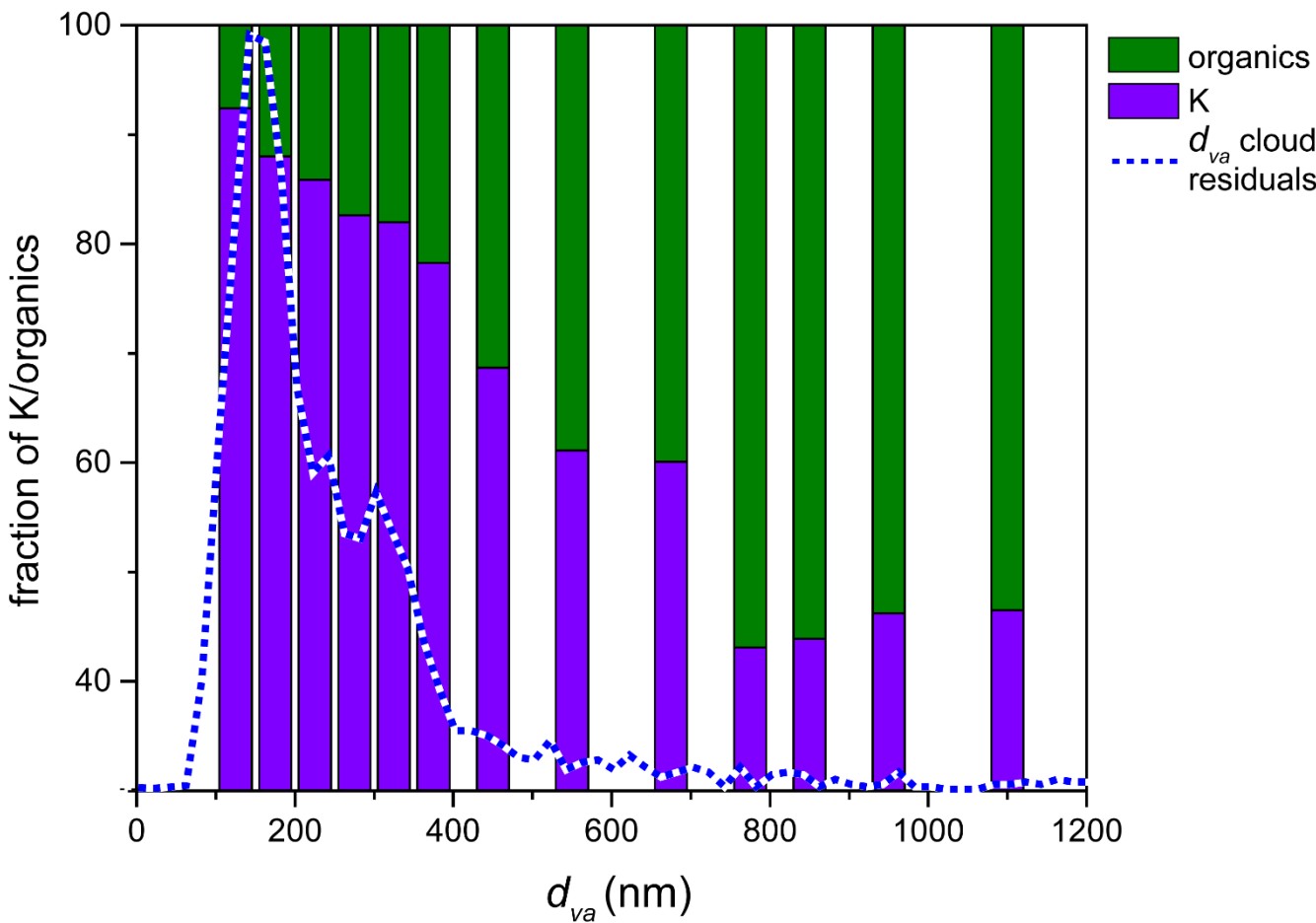

**Figure 12: The relative mass spectral intensity of peaks assigned to organics and potassium as a function of particle $d_{va}$, showing that larger particles contain more organics for PF CGina bacteria. The dashed line represents the measured $d_{va}$ distribution of cloud residuals.**

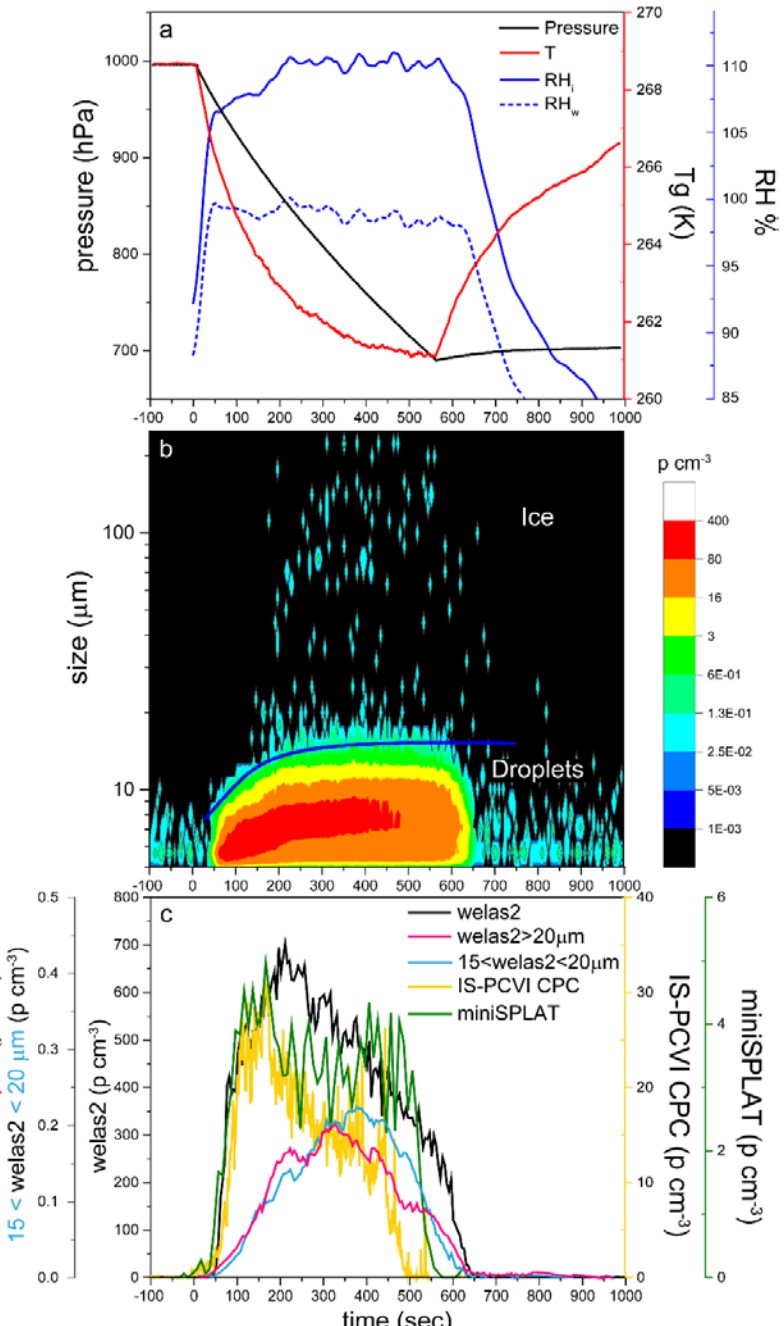

**Figure 13: Data from Expansion 3 with PF CGina bacteria. (a) The pressure (black) and the temperature of the gas (red) inside the AIDA chamber. The measured RH with respect to ice (RHi) and water (RHw) are indicated in solid and dashed blue lines, respectively; (b) the size distributions of the particles measured with welas2 during the expansion. Particles smaller than ~15 μm are liquid droplets, while larger particles are ice. The IS-PCVI cut-size is shown in blue; (c) The welas2-measured total particle number concentrations (black), the welas-2 number concentrations of particles larger than 15 μm, but smaller than 20 μm (light blue), and the welas-2 number concentrations of particles larger than 20 μm (pink). The CPC and miniSPLAT measured number concentrations of particles transmitted by the IS-PCVI are marked in yellow and green, respectively.**

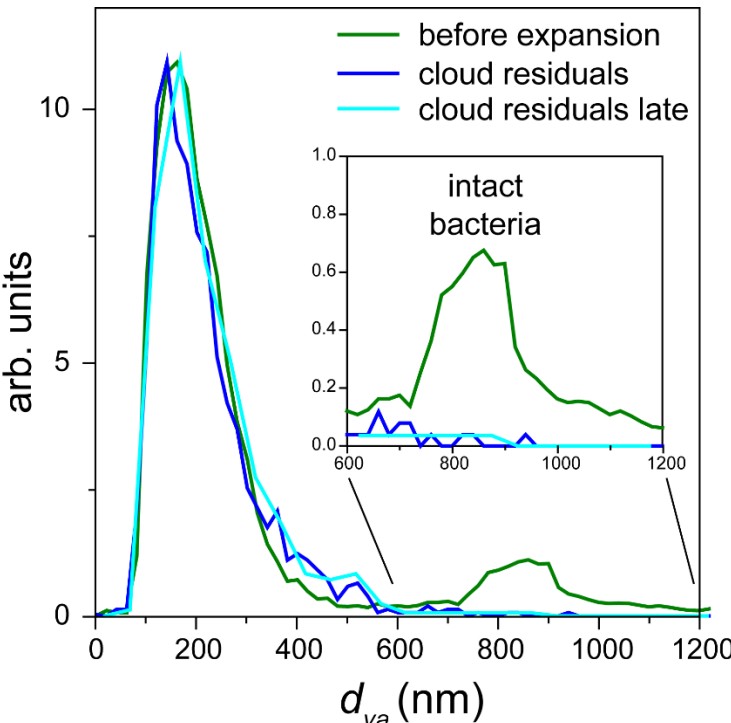

**Figure 14:** $d_{va}$ **size distribution measured by miniSPLAT before Expansion 3 (green) with PF CGina bacteria, of cloud residuals (blue), and of cloud residuals measured during the later part of the expansion (302-557 seconds) when the number concentration of sampled droplets and ice crystals were nearly the same (light blue). The inset shows an expanded scale of the region of intact bacteria.**

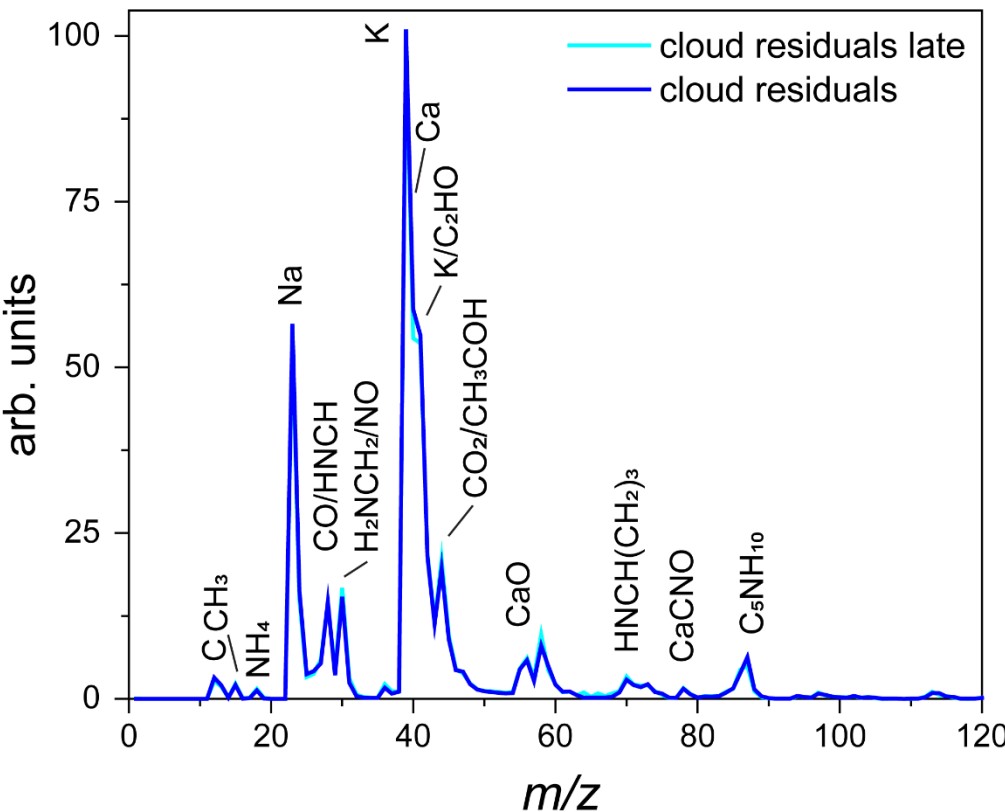

**Figure 15: Average mass spectrum of cloud residuals measured during Expansion 3 (blue) with PF CGina bacteria super imposed on the average mass spectrum of particles characterized later in the expansion (302-557 seconds) when the number of sampled ice crystals and cloud droplets were comparable (light blue).**