# Peer review of "Activation of Intact Bacteria and Bacterial Fragments Mixed with Agar as Cloud Droplets and Ice Crystals in Cloud Chamber Experiments"

_Atmospheric Chemistry and Physics, 2018_

## Referee Comment (RC1) · Anonymous Referee #1 · 12 Jul 2018

Suski et al present single-particle mass spectrometry (SPMS) measurements of bacteria and fragments that served as CCN and INPs in the AIDA cloud chamber during immersion cloud freezing experiments. This work tackles the important question of the role of primary biological particles in cloud formation and properties through laboratory experiments. I have questions below regarding interpretation of the data that may impact the results. Otherwise, revisions are recommended below primarily to increase clarity of the manuscript.

The main finding of this work is that bacteria fragments (mixed with agar) serve as cloud droplet nuclei, whereas intact bacteria rarely nucleate cloud droplets. The miniSPLAT

[Figure]

size distributions of particles prior to cloud formation (expansion) within the chamber compared to the cloud droplet residuals support this conclusion. However, I have several questions that may impact the interpretation of the results. 1) Could intact bacteria burst upon droplet activation or freezing? (Is there any support for this from previous studies?) Or, could intact bacteria burst during drying within the CVI? 2) What are the size cuts of the PCVI and IS-PCVI and transmission efficiencies? This information needs to be provided in the experimental section. For each expansion, what fraction of the droplets and ice crystals were and were not transmitted through the CVIs? (Provide in results & discussion.) 3) It is stated that the pressure changes during expansions impacted the miniSPLAT aerosol velocities. What was the dependence of the size-dependent miniSPLAT inlet transmission efficiency on these pressure changes during expansions? If particles >0.5 um were not as efficiently transmitted through the aero-dynamic lens at these lower pressures, this would explain why intact bacteria were not observed in the cloud particle residues.

Additional Major Comments: - Experimental (Page 5): Please provide additional information about the miniSPLAT operation. What was the LDI power, and did it vary during the study (since ion fragmentation is dependent on this)? What was the size-dependent inlet transmission efficiency as a function of inlet pressure? How were size calibrations conducted (PSLs?) and over what particle size range? What is the size range of efficient inlet transmission, and does it depend on pressure? (Figure 1 only gives a lower detection limit – no upper limit is provided, although it is clear in this figure that transmission appears to drop off above 1 um.) How were number concentrations obtained from the miniSPLAT data (what calibration/data processing was done)?

- Page 5, Line 31: In addition to different relative peak intensities, there appear to be differences in the individual ions present at > m/z 50. This should be discussed.

- To aid interpretation of the mass spectra, please label all ions in Figures 2, 5, 7, 10, and 14. If the chemical identity is not known, please list possibilities and/or at least the numerical m/z. Also, please label in the captions whether these mass spectra

correspond to averages or representative examples.

- Figure S1 is a very informative figure, and the authors may consider moving this to the main text. In particular, the labeling of the ions is useful, and the magnification of the small peaks is helpful. Regarding the comparison here, what differences are observed between the 'agar + bacterial fragments' and 'agar'? This should be discussed in the main text to support the inclusion of agar in the aerosols (Page 6, Line 8). It appears that the main difference may be the presence of m/z 131 in the bacterial fragments. Is this observed in the aerosols and cloud residues? The mass spectra shown in the main text are only shown up to m/z 120, so the reader cannot evaluate this. If m/z 131 is indeed a primary difference between the mass spectra, then I recommend that the authors show the mass spectra in the main text up to at least m/z 135.

- Page 6, Line 3: It is stated here that "intact cells show relatively lower intensities for the metal ions (23Na+, 39/41K+, 40Ca+)", but I do not see this in Figure 2.

- Page 6, Line 4: Please discuss the organic ions present, as they are not stated here in terms of m/z or ion formula, making interpretation of this statement by the reader not possible. There are many previous SPMS papers on primary biological particles that could aid in this mass spectral interpretation. A greater interpretation of the ions above m/z 41 may aid in the interpretation of the results. Example manuscripts to consult for SPMS biological particle mass spectra include (but are not limited to): Fergenson et al 2004 (Analytical Chem), Czerwienec et al 2005 (Analytical Chem), Srivastava et al 2005 (Analytical Chem). Peaks unique to the bacteria (fragments and intact) and not agar should be highlighted in this discussion.

- Page 7, Lines 23-25: It is stated here that "The small differences between the MS of residuals and the small particle mode suggest a slightly higher content of organics in the residuals", but I do not see this in Figure 5. Please provide additional support/description (e.g. label and discuss specific peaks).

- Page 9, Line 3: This sentence implies that peaks > m/z 44 are all organics, which

contradicts Page 6, Lines 1-2 and previous SPMS biological particle studies that show higher mass inorganic peaks as well. This discussion should consider specific peaks and fix this statement.

- Page 9, Lines 28-30: Typically CVIs concentrate cloud residuals during sampling and this transmission is size-dependent. This information needs to be stated in the experimental section, and this needs to be considered in the interpretation of the number concentrations measured after the CVI, as discussed on this page. The reader also does not have knowledge of the transmission differences between the IS-PCVI and PCVI to evaluate the transmission of cloud droplets vs ice crystals as stated here.

- Page 11, Line 31 – Page 12, Lines 1-2: This result was not shown or discussed in the results & discussion, and if included here in the conclusions, it should be shown and discussed.

- To aid in comparison of the results between the different expansions, the authors are encouraged to combine (into a multi-panel plot or by stacking the mass spectra) the following mass spectral figures: 5, 7, 10, and 14. Similarly, interpretation would be improved with combining the following size distribution figures: 4, 9, and 13. Figures 1 and 6 could be similarly combined.

- Figure 11 and associated discussion: Since ion intensities in laser desorption ionization are dependent on ionization energies (e.g. Gross et al 2000, Analytical Chem.; Reinard & Johnston 2008, J. ASMS), the ion ratios used here need to be discussed in greater detail as they do not represent quantitative mole or mass ratios. In particular, inorganic ions typically have much lower ionization efficiencies compared to organic ions, which also undergo significant fragmentation at 193 nm. In addition, the organic ions included here in the ratio need to be stated here, in the experimental, or in the results & discussion. While it is clear that the organic contribution increases with size (and this is a useful result), clarification needs to be provided for the reader not familiar with LDI. Additionally, the text on page 9 refers to the dva distribution here as a

'schematic representation'. Please clarify in the caption and show actual data instead.

Minor Comments: - Introduction: Many sentences do not include references, which should be added to support the statements. In particular, references are needed on page 2, lines 1, 12, 14, 20, and 21.

- Page 3, Line 23: Move this sentence to the end of the previous paragraph, as one sentence does not constitute a full paragraph.

- Page 4: RH is defined twice here as well as on page 2.

- Page 4, line 15: Why are the SMPS and APS data converted to volume-equivalent diameters? It would seem more appropriate, for comparison to the miniSPLAT data (in vacuum aerodynamic diameter), to simply convert the SMPS mobility diameters to aerodynamic diameter and leave the APS data in aerodynamic diameter. Also, please provide a reference for the chosen particle density.

- Results & Discussion: It would be useful for the reader for sub-headers to be added, for example, 1) Bacteria mass spectral signatures, 2) Expansion 1, 3) Expansion 2. . ..

- Page 5, Line 18: Space needed after (dve).

- Page 6, Lines 11-13: This sentence is not clear as written. Laser power should be provided in the experimental section.

- Page 7, Lines 26-27: Move this sentence to later in the results & discussion where Expansion 2 is discussed.

- Page 7, Last paragraph: This paragraph is all repetition (summary) and could be removed as it doesn't seem necessary.

- Page 9, Lines 7-10: Please clarify this sentence.

- Page 9, Line 19: Should Figure 13 be referred to here instead of Figure 6a (typo?)?

- Page 9, Line 24: Add mention of the number concentration prior to expansion for

context.

- Page 10, Lines 3-4: Move "(light blue)" to after "expansion" in this sentence.

- Page 10, Lines 10-12: Move sentence to the end of the previous paragraph.

- Page 11, Line 23: Please provide a reference for the hygroscopicity of intact bacteria.

- Figure 9 caption: Mention the type of bacteria used. Make sure this is included in each figure caption.

- Figure 12: Please clarify the legend of the bottom plot.

- Figure 13: Please provide the sample timing for the 'cloud residuals' vs 'cloud residuals late', so that the reader can refer back to Figure 12 for context.

---

## Referee Comment (RC2) · Anonymous Referee #2 · 25 Aug 2018

Suski et al. present experimental data from three cloud chamber expansions at AIDA where a suspension of bacterial particles and their fragments mixed with agar were injected into the chamber. The goal appears to be to understand and contrast the role of intact bacterial cells versus bacterial fragments mixed with agar in the nucleation of cloud droplets than can then undergo immersion freezing. This is certainly a relevant question to the atmospheric science community, though the presence of agar makes the results more relevant to interpreting past and future laboratory studies that use neb-ulized suspensions of bacterial since agar is not an atmospherically relevant particle component. In the end few new findings are really presented from these experiments, and the significance and originality of this work is rather low as a result. Perhaps the

most novel aspect is the use of a cloud expansion chamber. I also found the paper hard to follow, and do not think the very narrative style of discussing each of the three cloud expansions to be an effective way to communicate, analyze, and synthesize the results. The main finding is that intact bacterial particles make a small contribution to CCN activation and thus cloud droplets, and thus also a small contribution to immersion freezing and ice crystal production. This is certainly a worthwhile finding and it should really be made the focus of this paper, but it also requires better support from the available data. I was not very convinced by the interpretation of these results, especially since the intact bacterial were such a small contribution to the total particle numbers to begin with. The single-particle mass spectrometer SPLAT is used to determine the chemical composition as a function of size. This analysis is hampered by the lack of significantly distinct mass spectral features that can be used to reliably distinguish the fragment+agar particles from the intact bacteria. As it is a single-particle instrument, as droplet and ice crystal activation occurs on individual particles, I found it odd that average mass spectra were presented, as opposed to trying to determine the fraction of each type of particle as a function of particle size. In summary, while the topic is of interest, not much new insight is presented here, and the presentation and discussion is quite unclear and hard to follow. The main singular conclusion that intact bacteria do not activate as CCN or into ice crystals needs further support. Extensive revisions are required to achieve this, and publication in ACP may not be warranted unless these major issues can be properly resolved.

I find the narrative style of describing each expansion experimental chronologically to be an ineffective way to communicate the results. At the least the important characteristics of each expansion and how they differ from each other must be discussed. E.g. how do the aerosol concentrations and size distributions differ? How do the thermodynamic conditions of the expansion differ? It looks like expansion 1 reaches a higher supersaturation of at least 102% RHw. Stating the maximum SS/RHw reached in each expansion is critical for understanding the CCN activation, just as stating the maximum RHi and minimum temperature reached is critical for understanding the ice

crystal production.

In describing each expansion the authors mostly state what data is plotted in the various figures, as opposed to actually discussing, interpreting, and synthesizing the results. As written the description of the expansion experiments is not very meaningful.

On "hydrophobic" intact bacterial particles. So long as the particle surface is wettable these large particles will still activate under the high supersaturations reached here. It appears that RHw usually hits 101%, so 1% SS. This is why it is important to state the maximum RHw. At high a high SS large particles with a very small hygroscopicity of kappa ∼ 0 will still activate into droplets.

What would really help this analysis is if the /fraction/ of the two type types/size modes of particles that activate into cloud droplets and ice crystals in each expansion could be estimated. This should be possible from the data. It is rather misleading to say that the intact bacteria make a small contribution to cloud droplets and ice crystals considering they are a small fraction of the initial particles to begin with. What is needed are the particle fractions, ie the CCN/CN and INP/CN ratios. This will also make these results more useful in a quantitative manner for other researchers. As presented the results reported here cannot be used in a meaningful to for example describe the CCN or IN properties of these particles types in a model.

Pg 1/line 30: Please provide several references for this b/g info on the role of bacteria in the atmosphere and clouds. "Murray, 2012 and references therein" is not sufficient. One suggestion:

DeMott, P. J. and Prenni, A. J.: New Directions: Need for defining the numbers and sources of biological aerosols acting as ice nuclei, Atmos. Environ., 44(15), 1944–1945, doi:10.1016/j.atmosenv.2010.02.032, 2010.

2/7: Contact angle would explain only the wettablilty, not hygroscopicity, of bacteria. Hygroscopicity really refers to the ability of a dissolved solute solution to take up water.

2/15: The introduction is quite spare on references to the many prior papers on the ice nucleation properties of bacteria/Pseudomonas syringae/Snomax. Some of these get mentioned much later in the paper but they also belong in the introduction, or else it appears that the authors are not familiar enough with the topic under study here. Some suggestions, but there are many more:

Pandey, R., Usui, K., Livingstone, R. A., Fischer, S. A., Pfaendtner, J., Backus, E. H. G., Nagata, Y., Fro hlich-Nowoisky, J., Schmu ser, L., Mauri, S., Scheel, J. F., Knopf, D. A., Po schl, U., Bonn, M. and Weidner, T.: Ice-nucleating bacteria control the order and dynamics of interfacial water, Sci. Adv., 2(4), e1501630–e1501630, doi:10.1126/sciadv.1501630, 2016.

Lindow, S. E., Arny, D. C. and Upper, C. D.: Bacterial Ice Nucleation: A Factor in Frost Injury to Plants, PLANT Physiol., 70(4), 1084–1089, doi:10.1104/pp.70.4.1084, 1982.

Després, V., Huffman, J. A., Burrows, S. M., Hoose, C., Safatov, A., Buryak, G., Fröhlich-Nowoisky, J., Elbert, W., Andreae, M., Pöschl, U. and Jaenicke, R.: Primary biological aerosol particles in the atmosphere: a review, Tellus B Chem. Phys. Meteorol., 64(1), 15598, doi:10.3402/tellusb.v64i0.15598, 2012.

Polen, M., Lawlis, E. and Sullivan, R. C.: The unstable ice nucleation properties of Snomax[®] bacterial particles, J. Geophys. Res. Atmos., 121(19), 11,666-11,678, doi:10.1002/2016JD025251, 2016.

Wex, H., Augustin-Bauditz, S., Boose, Y., Budke, C., Curtius, J., Diehl, K., Dreyer, A., Frank, F., Hartmann, S., Hiranuma, N., Jantsch, E., Kanji, Z. A., Kiselev, A., Koop, T., Möhler, O., Niedermeier, D., Nillius, B., Rösch, M., Rose, D., Schmidt, C., Steinke, I. and Stratmann, F.: Intercomparing different devices for the investigation of ice nucleating particles using Snomax[®] as test substance, Atmos. Chem. Phys., 15(3), 1463–1485, doi:10.5194/acp-15-1463-2015, 2015.

Pummer, B. G., Bauer, H., Bernardi, J., Bleicher, S. and Grothe, H.: Suspendable

macromolecules are responsible for ice nucleation activity of birch and conifer pollen, Atmos. Chem. Phys., 12(5), 2541–2550, doi:10.5194/acp-12-2541-2012, 2012.

Turner, M. A., Arellano, F. and Kozloff, L. M.: Components of ice nucleation structures of bacteria, J. Bacteriol., 173(20), 6515–6527, 1991.

Turner, M. A., Arellano, F. and Kozloff, L. M.: Three separate classes of bacterial ice nucleation structures, J. Bacteriol., 172(5), 2521–2526, 1990.

Attard, E., Yang, H., Delort, A.-M., Amato, P., Pöschl, U., Glaux, C., Koop, T. and Morris, C. E.: Effects of atmospheric conditions on ice nucleation activity of <i>Pseudomonas</i>, Atmos. Chem. Phys., 12(22), 10667–10677, doi:10.5194/acp-12-10667-2012, 2012.

Hartmann, S., Augustin, S., Clauss, T., Voigtländer, J., Niedermeier, D., Wex, H. and Stratmann, F.: Immersion freezing of ice nucleating active protein complexes, Atmos. Chem. Phys. Discuss., 12(8), 21321–21353, doi:10.5194/acpd-12-21321-2012, 2012.

Vali, G., Christensen, M., Fresh, R. W., Galyan, E. L., Maki, L. R. and Schnell, R. C.: Biogenic ice nuclei 2. Bacterial sources, J. Atmos. Sci., 33(8), 1565–1570, 1976.

2/23: Macromolecules of protein aggregates are known ice nucleants produced by bacteria. Are the macromolecules really a "solid nucleus". This definition doesn't align with the role of macromolecules as ice nucleants. 2/26: Consider using "macromolecules" instead of "nano-INP", as this is the terminology more widely used.

3/3: There is also evidence of biological ice nucleating macromolecules attaching to particles such as dust, and also evidence for mixed "dust-bio" particles, such as from the first author's prior work. This would also seem to motivate this study and should be discussed.

O'Sullivan, D., Murray, B. J., Ross, J. F. and Webb, M. E.: The adsorption of fungal ice-nucleating proteins on mineral dusts: a terrestrial reservoir of atmospheric ice-nucleating particles, Atmos. Chem. Phys., 16(12), 7879–7887, doi:10.5194/acp-

16-7879-2016, 2016.

3/27: It is a bit confusing that the stated purpose of FIN-1 is to intercompare SP-MS instruments and yet that is not done here. Please explain more to better put this particular study in the context of FIN-1.

4/14: Usually the particle density and shape are varied to arrive at a good overlap in the SMPS and APS size distributions, instead of just assuming values. Also what are these assumed values based on? Isn't the SPLAT instrument a great way to actually measure the shape factor and density of these particles?

Khlystov, A., Stanier, C. and Pandis, S. N.: An Algorithm for Combining Electrical Mobility and Aerodynamic Size Distributions Data when Measuring Ambient Aerosol, Aerosol Sci. Technol., 38(sup1), 229–238, doi:10.1080/02786820390229543, 2004.

Beddows, D. C. S., Dall'osto, M. and Harrison, R. M.: An Enhanced Procedure for the Merging of Atmospheric Particle Size Distribution Data Measured Using Electrical Mobility and Time-of-Flight Analysers, Aerosol Sci. Technol., 44(11), 930–938, doi:10.1080/02786826.2010.502159, 2010.

4/27: Please state the cut-size of these two CVIs.

5/10: Why were these two cultures chosen? Much existing work of course on P. syringae, but what about PF CGina 01? Also unclear if these are both used in all three expansions?

6/15: Do you determine which particles are intact bacteria simply based on a size threshold, and if so what is it and what is it based on?

7/6: Qualitative terms such as "very high CCN activation efficiency" are often used. The hygroscopicity of the different aerosol types should be estimated from the maximum supersaturation observed, and the size distribution.

7/14: Why would these large bacterial particle not activate? Again need to know the

maximum in the RHw. Have you tested that these large particles once activated into cloud droplets survive the PCVI? I agree with the other referee's comments regarding the bacterial possibly rupturing during cloud droplet activation and certainly during freezing. Showing the average mass spectra is not that meaningful here. The small but important number fraction of intact bacteria will be obscured by averaging. An estimate of the number fraction of intact bacteria vs. fragments+agar in the different size modes would be really useful.

9/14: Again, need to know what the CCN/CN fraction was at what max RHw to really evaluate the hygroscopicity of the particles. "very high CCN activation efficiency" is not quantitative.

Expansion 3 seems unique in that there is much more ice produced even though the aerosol seems similar to the other expansions, but the reason for this difference is not discussed.

10/14: "Nevertheless, the data presented here show that bacterial fragments mixed with agar preferentially activate as droplets and are the only particles observed in ice residuals." This is essentially the singular conclusion reached from this study, and it is an interesting one. I strongly suggest making this aspect the focus and providing more data and analysis that better supports this conclusion.

Fig. 2 and other mass spectra: The two types of particles appear to have no unique ion markers, just differences in the relative ion signals. This makes separating the two particle types out based on their mass spectra rather challenging.

---

## Author Comment (AC1) · 15 Oct 2018

Response to Anonymous Referee #1

Suski et al present single-particle mass spectrometry (SPMS) measurements of bacteria and fragments that served as CCN and INPs in the AIDA cloud chamber during immersion cloud freezing experiments. This work tackles the important question of the role of primary biological particles in cloud formation and properties through laboratory experiments. I have questions below regarding interpretation of the data that may impact the results. Otherwise, revisions are recommended below primarily to increase clarity of the manuscript. The main finding of this work is that bacteria fragments

(mixed with agar) serve as cloud droplet nuclei, whereas intact bacteria rarely nucleate cloud droplets. The miniSPLAT size distributions of particles prior to cloud formation (expansion) within the chamber compared to the cloud droplet residuals support this conclusion. However, I have several questions that may impact the interpretation of the results. Thank you for your detailed comments, suggestions, and questions. Below are our responses.

1) Could intact bacteria burst upon droplet activation or freezing? (Is there any support for this from previous studies?)

There is no evidence that bursting is occurring in the cloud chamber. If the intact cells were bursting during the expansion, they would not be observed post-expansion in roughly equal fractions as they were pre-expansion, as shown in the new Figure 7.

Or, could intact bacteria burst during drying within the CVI? The aerosolized bacteria were dried prior to injection into the AIDA chamber; therefore, if bursting of these specific intact bacterial cells were to occur during drying, it should happen during this initial drying process as well. Consequently, it is unlikely that bursting occurs upon drying in the PCVI or IS-PCVI. Moreover, if intact bacteria activate and then burst in the PCVI, we would expect to see an increase in the number of particles measured by the CPC that is downstream of the PCVI, similar to what is seen during aircraft flights when shattering of large droplets and ice crystals in CVIs occurs. These types of concentration spikes were not observed in the present study.

2) What are the size cuts of the PCVI and IS-PCVI and transmission efficiencies? This information needs to be provided in the experimental section. For each expansion, what fraction of the droplets and ice crystals were and were not transmitted through the CVIs? (Provide in results & discussion.)

The cut-sizes for the PCVI and IS-PCVI used in the 3 expansions have now been included on the expansion plots (Figures 4 9, and 13), as well as in Table S1 in the supplement. We have also calculated the fraction of droplets and ice crystals

[Figure]

that were transmitted using the equation below and have plotted the fractions in Figure S1. Fraction of the droplets and ice crystals reached at the CVI(t)=(PCVI CPC(t)×F_output/F_input )/(Welas2(t)) where Foutput is the CVI output flow (lpm), and Finput is the CVI input flow (lpm).

Table S1 also lists the transmission efficiencies of 94.4% for Expansion 1 and 61.9% for Expansion 2 determined according to Figure 3 of Gallavardin et al. (2008) and 91.5% for Expansion 3 determined according to Fig. 13 of Hiranuma et al. (2016).

Gallavardin, S. J., Froyd, K. D., Lohmann, U., Möhler, O., Murphy, D. M., and Cziczo, D. J.: Single particle laser mass spectrometry applied to differential ice nucleation experiments at the AIDA chamber, Aerosol Sci. Tech., 42, 773–791, doi:10.1080/02786820802339538, 2008.

Hiranuma, N., Möhler, O., Kulkarni, G., Schnaiter, M., Vogt, S., Vochezer, P., Jarvinen, E., Wagner, R., Bell, D. M., Wilson, J., Zelenyuk, A., and Cziczo, D. J.: Development and characterization of an ice-selecting pumped counterflow virtual impactor (IS-PCVI) to study ice crystal residuals, Atmos Meas Tech, 9, 3817-3836, 10.5194/amt-9-3817-2016, 2016.

3) It is stated that the pressure changes during expansions impacted the miniSPLAT aerosol velocities. What was the dependence of the size dependent miniSPLAT inlet transmission efficiency on these pressure changes during expansions? If particles >0.5 um were not as efficiently transmitted through the aerodynamic lens at these lower pressures, this would explain why intact bacteria were not observed in the cloud particle residues.

The size-dependent transmission efficiency of the aerodynamic lens inlet, used by the AMS and miniSPLAT, is indeed expected to change with pressure. Liu et al. 2007 presents the results of CFD modeling and experimental data for sampling pressures of 760 torr and 585 torr. It shows that while the model predicts a decreased transmission efficiency for larger particles, measurements on 3 different particle types show

virtually no change for the transmission efficiency of larger particles. The reference to this paper has also been added to the revised manuscript. Most importantly, our measurements conducted during other expansions in the AIDA chamber show that when larger particles activate more efficiently, it can be clearly observed in the miniSPLAT-measured dva distributions. For example, Figure 1 in the responses shows the normalized miniSPLAT-measured dva size distributions measured before and during an expansion with feldspar present in the chamber. The size distribution of cloud residuals sampled during the expansion clearly shows an enhancement in the relative fraction of larger particles. This suggests that larger particles are effectively transmitted through the PCVI and the miniSPLAT inlet during expansions when pressures are lower than normal atmospheric pressure, consistent with the data reported in Liu et al. 2007.

Peter S. K. Liu , Rensheng Deng , Kenneth A. Smith , Leah R. Williams , John T. Jayne , Manjula R. Canagaratna , Kori Moore , Timothy B. Onasch , Douglas R. Worsnop & Terry Deshler (2007) Transmission Efficiency of an Aerodynamic Focusing Lens System: Comparison of Model Calculations and Laboratory Measurements for the Aerodyne Aerosol Mass Spectrometer, Aerosol Science and Technology, 41:8, 721-733, DOI: 10.1080/02786820701422278

Additional Major Comments: - Experimental (Page 5): Please provide additional information about the miniSPLAT operation. What was the LDI power, and did it vary during the study (since ion fragmentation is dependent on this)?

The text has been modified to include this information. "The dual particle detection is also used to generate a trigger for the excimer laser (GAM Lasers Inc., Model EX-5), operated at 193 nm with a constant laser energy of $1.0 \pm 0.1$ mJ/pulse, which ablates the particles and generates positive and negative ions."

What was the size dependent inlet transmission efficiency as a function of inlet pressure?

A full size-dependent transmission efficiency curve was not generated for each inlet

pressure, but as discussed above, larger particles >0.5 um were transmitted during expansions. Figure 1 shows that the ratio of the two particle types measured by miniSPLAT before the expansion is consistent with the size distributions generated from the SMPS and APS. The important issue here is the difference in the ratio of the concentrations of the two particle types before the expansion and of cloud residuals during the expansion. Size-dependent transmission efficiency curves for two pressures and 3 particle types are given in Liu et al. 2007 (Figures 9 – 11, Table 3). Using the data presented by Liu et al. in Table 3 we plotted their measured averaged transmission efficiencies for two pressures shown in Figure 2 in the response. The miniSPLAT size distributions measured before and after the expansions for the 3 experiments show two clear peaks for two particle types, while the cloud residual size distributions show that the large particle mode is missing. It is not that no large particles were detected in cloud residuals, they are clearly detected for other samples as explained above, but the distinct peak of the intact bacteria mode is not present.

Peter S. K. Liu , Rensheng Deng , Kenneth A. Smith , Leah R. Williams , John T. Jayne , Manjula R. Canagaratna , Kori Moore , Timothy B. Onasch , Douglas R. Worsnop & Terry Deshler (2007) Transmission Efficiency of an Aerodynamic Focusing Lens System: Comparison of Model Calculations and Laboratory Measurements for the Aerodyne Aerosol Mass Spectrometer, Aerosol Science and Technology, 41:8, 721-733, DOI: 10.1080/02786820701422278

How were size calibrations conducted (PSLs?) and over what particle size range?

This has been added to the text. "Given that the inlet pressure is changing during the expansions, polystyrene latex spheres (PSLs) of known sizes ranging from 73 nm to 993 $\mu$m were used to generate a pressure dependent size calibration curve over a range of pressures from 0.9 to 4 Torr."

What is the size range of efficient inlet transmission, and does it depend on pressure? (Figure 1 only gives a lower detection limit – no upper limit is provided, although it is

clear in this figure that transmission appears to drop off above 1 um.)

The size-dependent transmission efficiency of the inlet used in the SPLAT II and miniS-PLAT instruments at normal atmospheric pressure is provided in Zelenyuk et al. (2009). Liu at al. (2007) reports measured size-dependent transmission efficiencies at 2 sampling pressures (Response Figure 2). Our measurements conducted during other expansions in the AIDA chamber show that ∼micron-sized particles like feldspar that activated as CCN and/or IN are transmitted and detected by miniSPLAT.

Zelenyuk, A., Yang, J., Choi, E., Imre, D. (2009). SPLAT II: An Aircraft Compatible, Ultra-Sensitive, High Precision Instrument for In-Situ Characterization of the Size and Composition of Fine and Ultrafine Particles. Aerosol Science and Technology 43:411-424.

Peter S. K. Liu , Rensheng Deng , Kenneth A. Smith , Leah R. Williams , John T. Jayne , Manjula R. Canagaratna , Kori Moore , Timothy B. Onasch , Douglas R. Worsnop & Terry Deshler (2007) Transmission Efficiency of an Aerodynamic Focusing Lens System: Comparison of Model Calculations and Laboratory Measurements for the Aerodyne Aerosol Mass Spectrometer, Aerosol Science and Technology, 41:8, 721-733, DOI: 10.1080/02786820701422278

How were number concentrations obtained from the miniSPLAT data (what calibration/data processing was done)?

This has been added to the text for clarity. "miniSPLAT-measured particle number concentrations are calculated by dividing the particle detection rate at the first optical detection stage by the pressure-dependent sampling flow rate as described in detail in Vaden et al. (2011a)." This approach was previously applied to aircraft-based measurements and resulted in an average difference of 0.5% between 1-sec concentrations measured by miniSPLAT and a dedicated particle counter (Vaden et al. (2011a)).

- Page 5, Line 31: In addition to different relative peak intensities, there appear to be

differences in the individual ions present at > m/z 50. This should be discussed.

It is easier to compare the mass spectra of the two particle modes using the new Figure 3 (original Figure S1), in which the lower intensity peaks are magnified. It shows that the mass spectra of the two particle types in the chamber have the same mass spectral peaks, just with different relative intensities.

- To aid interpretation of the mass spectra, please label all ions in Figures 2, 5, 7, 10, and 14. If the chemical identity is not known, please list possibilities and/or at least the numerical m/z.

In the new version of the paper these figures include labeled mass spectral peaks.

Also, please label in the captions whether these mass spectra correspond to averages or representative examples.

They are averages. A note was added to the figure captions.

- Figure S1 is a very informative figure, and the authors may consider moving this to the main text. In particular, the labeling of the ions is useful, and the magnification of the small peaks is helpful. Regarding the comparison here, what differences are observed between the 'agar + bacterial fragments' and 'agar'? This should be discussed in the main text to support the inclusion of agar in the aerosols (Page 6, Line 8). It appears that the main difference may be the presence of m/z 131 in the bacterial fragments. Is this observed in the aerosols and cloud residues? The mass spectra shown in the main text are only shown up to m/z 120, so the reader cannot evaluate this. If m/z 131 is indeed a primary difference between the mass spectra, then I recommend that the authors show the mass spectra in the main text up to at least m/z 135.

We have moved this figure to the main text. Both aerosolized intact bacteria and bacteria fragments mixed with agar are composed of the same compounds, albeit present in different ratios. Therefore, it is not surprising that the mass spectra of all particles in the chamber have the same mass spectral peaks just with different relative intensities, as shown in the new Figure 3. All of these particles, however, contain distinct phosphorus-containing peaks that separate them from pure agar. The peak at m/z 131 (possibly, C7H17NO) also appears to be characteristic of bacteria and of bacterial fragments. Additionally, one of the findings presented in this manuscript is that the fraction of agar and bacterial fragments is changing with particle size. Figure 12 of the revised manuscript and Figure 3 in the response clearly illustrate this point. The simultaneous measurements of single particle composition and size provides the means to distinguish between the two particle types. However, distinguishing between intact bacteria and large particles composed of bacterial fragments mixed with agar based on their mass spectra is impossible.

- Page 6, Line 3: It is stated here that "intact cells show relatively lower intensities for the metal ions (23Na+, 39/41K+, 40Ca+)", but I do not see this in Figure 2.

The mass spectra shown in Figure 2 are normalized to the most intense peak (K+). The intensity of this peak for intact bacteria represents a much lower fraction of the total mass spectral intensity (sum of all red peaks) as compared to the bacteria fragments mixed with agar. Figure 3 in the response shows the size-dependent mass spectra of bacterial fragments mixed with agar, which were normalized by the total MS intensity. It clearly shows that the relative intensity of K+ decreases with particle size. Similarly, Figure 12, which used to be Figure 11 in the original version, provides another display of the decreasing relative intensity of the K peak with particle size.

- Page 6, Line 4: Please discuss the organic ions present, as they are not stated here in terms of m/z or ion formula, making interpretation of this statement by the reader not possible. There are many previous SPMS papers on primary biological particles that could aid in this mass spectral interpretation. A greater interpretation of the ions above m/z 41 may aid in the interpretation of the results. Example manuscripts to consult for SPMS biological particle mass spectra include (but are not limited to): Fergenson et al 2004 (Analytical Chem), Czerwienec et al 2005 (Analytical Chem), Srivastava et al 2005 (Analytical Chem). Peaks unique to the bacteria (fragments and intact) and not

agar should be highlighted in this discussion.

We have expanded this discussion to include more ion identification.

- Page 7, Lines 23-25: It is stated here that "The small differences between the MS of residuals and the small particle mode suggest a slightly higher content of organics in the residuals", but I do not see this in Figure 5. Please provide additional support/description (e.g. label and discuss specific peaks).

We have added more information about the organic ions in the mass spectra and expanded the discussion.

- Page 9, Line 3: This sentence implies that peaks > m/z 44 are all organics, which contradicts Page 6, Lines 1-2 and previous SPMS biological particle studies that show higher mass inorganic peaks as well. This discussion should consider specific peaks and fix this statement.

We have revised this discussion and modified the original Figure 11 (new Figure 12). There are a number of ways to illustrate that the K+ peak intensity relative to organic peaks decreases with particle size. Figure 12 and Figure 3 in the response, all illustrate this important point. One of the important findings presented in this paper is that the relative concentrations of agar and bacterial fragments in the small particle mode change with particle size, such that the fraction of agar decreases with particle size.

- Page 9, Lines 28-30: Typically CVIs concentrate cloud residuals during sampling and this transmission is size-dependent. This information needs to be stated in the experimental section, and this needs to be considered in the interpretation of the number concentrations measured after the CVI, as discussed on this page. The reader also does not have knowledge of the transmission differences between the IS-PCVI and PCVI to evaluate the transmission of cloud droplets vs ice crystals as stated here.

The transmission efficiencies and cut-sizes have been added to the text. The size dependent concertation factor would be based on droplet or ice crystal size and not

residual size. Therefore, calculating the size dependent residual enhancement is not relevant. Average particle concentration enhancement factor is calculated according to Eqn. 6 of Hiranuma et al. (2016) and was 2 for the PCVI and 12 for the IS-PCVI. This information has also been added to the text and the table in the supplement.

- Page 11, Line 31 – Page 12, Lines 1-2: This result was not shown or discussed in the results & discussion, and if included here in the conclusions, it should be shown and discussed.

This result is from unpublished data, so we have added a reference to a personal communication for this result. The ns value for agar is 7.82 x 108 m-2 at -22.95 °C (250.20 K).

- To aid in comparison of the results between the different expansions, the authors are encouraged to combine (into a multi-panel plot or by stacking the mass spectra) the following mass spectral figures: 5, 7, 10, and 14. Similarly, interpretation would be improved with combining the following size distribution figures: 4, 9, and 13. Figures 1 and 6 could be similarly combined.

We find that data in combined multi-panel plots are difficult to comprehend and interpret. Most importantly, we chose to present the results as three separate experiments to emphasize that the data from three separate experiments and two different bacteria are perfectly reproducible.

- Figure 11 and associated discussion: Since ion intensities in laser desorption ionization are dependent on ionization energies (e.g. Gross et al 2000, Analytical Chem.; Reinard & Johnston 2008, J. ASMS), the ion ratios used here need to be discussed in greater detail as they do not represent quantitative mole or mass ratios. In particular, inorganic ions typically have much lower ionization efficiencies compared to organic ions, which also undergo significant fragmentation at 193 nm. In addition, the organic ions included here in the ratio need to be stated here, in the experimental, or in the results & discussion. While it is clear that the organic contribution increases with size

(and this is a useful result), clarification needs to be provided for the reader not familiar with LDI. Additionally, the text on page 9 refers to the dva distribution here as a 'schematic representation'. Please clarify in the caption and show actual data instead.

The revised figure (new Figure 12) shows the actual dva distribution. As we discussed above, there are a number of ways to illustrate that the intensity of the K+ peak relative to organic peaks decreases with particle size. Figure 12 and Figure 3 in the response, all illustrate this important point. The text also clearly states that the ratios presented in Figure 12 serve as a simple qualitative measure of the relative fraction of bacterial fragments and agar in these particles. We have modified the text to clarify how the ratio shown in the figure was calculated. We agree that this is one of the important findings in this study.

Minor Comments: - Introduction: Many sentences do not include references, which should be added to support the statements. In particular, references are needed on page 2, lines 1, 12, 14, 20, and 21.

References have been added to these sentences.

- Page 3, Line 23: Move this sentence to the end of the previous paragraph, as one sentence does not constitute a full paragraph.

It has been moved.

- Page 4: RH is defined twice here as well as on page 2.

This has been fixed. Thank you.

- Page 4, line 15: Why are the SMPS and APS data converted to volume-equivalent diameters? It would seem more appropriate, for comparison to the miniSPLAT data (in vacuum aerodynamic diameter), to simply convert the SMPS mobility diameters to aerodynamic diameter and leave the APS data in aerodynamic diameter. Also, please provide a reference for the chosen particle density.

A similar application of volume equivalent diameter in SMPS and APS data for other studies in the AIDA chamber can be found in Hiranuma et al 2015 (Nature Geosci.; DOI: 10.1038/NGEO2374, See Supplemental Information Section 1.3) and many other publications. In addition, the volume-equivalent diameter was used for the SMPS/APS data to be comparable to Welas2-measured metrics (e.g., Hiranuma et al., 2015; Atmos. Chem. Phys., 15, 2489–2518, 2015). Sect. 3.4 of Hiranuma et al. (2016) and references therein describe why volume-equivalent diameter is the best choice for displaying welas2 data, but briefly it is because we do not know the geometry of the ice crystals. The density of 1.4 g/cm3 was used here to arrive at a good overlap in the SMPS and APS size distributions, assuming particle sphericity (Peters et al. (2006)). We have added more clarification on this in the text. The effective density of this bacteria of 1.4 g cm-3 has been previously measured in Wex et al. 2015 and is in a good agreement with the effective density of 1.38 g cm-3 measured by miniSPLAT for intact pseudomonas syringae during the FIN-1 campaign. Moreover, the vacuum aerodynamic diameter measured by miniSPLAT is not the same as aerodynamic diameter measured by APS. What is important for this study is that 2 distinct particle modes are clearly visible in the miniSPLAT and SMPS/APS-measured size distributions.

Hiranuma, N., Möhler, O., Kulkarni, G., Schnaiter, M., Vogt, S., Vochezer, P., Jarvinen, E., Wagner, R., Bell, D. M., Wilson, J., Zelenyuk, A., and Cziczo, D. J.: Development and characterization of an ice-selecting pumped counterflow virtual impactor (IS-PCVI) to study ice crystal residuals, Atmos Meas Tech, 9, 3817-3836, 10.5194/amt-9-3817-2016, 2016.

Wex, H., Augustin-Bauditz, S., Boose, Y., Budke, C., Curtius, J., Diehl, K., Dreyer, A., Frank, F., Hartmann, S., Hiranuma, N., Jantsch, E., Kanji, Z. A., Kiselev, A., Koop, T., Möhler, O., Niedermeier, D., Nillius, B., Rosch, M., Rose, D., Schmidt, C., Steinke, I., and Stratmann, F.: Intercomparing different devices for the investigation of ice nucleating particles using Snomax (R) as test substance, Atmos Chem Phys, 15, 1463-1485, 10.5194/acp-15-1463-2015, 2015.

Peters, T. M., Ott, D. & O'Shaughnessy, P. T. Comparison of the Grimm 1.108 and 1.109 388 portable aerosol spectrometer to the TSI 3321 aerodynamic particle sizer for dry particles. 389 Annals of Occupational Hygiene, 50, 843–850, doi:10.1093/annhyg/mel067 (2006).

- Results & Discussion: It would be useful for the reader for sub-headers to be added, for example, 1) Bacteria mass spectral signatures, 2) Expansion 1, 3) Expansion 2. . ..

We have added sub-headers to the text.

- Page 5, Line 18: Space needed after (dve).

This has been fixed.

- Page 6, Lines 11-13: This sentence is not clear as written. Laser power should be provided in the experimental section.

This has been clarified as "The negative ion MS presented here and in previous studies are virtually the same, while the positive ion MS presented in this study exhibit significantly higher intensity in organic peaks, most likely due to the differences in the ablation laser power or wavelength (266 nm in Pratt et al. (2009) and 193 nm in this study and Zawadowicz et al. (2017)). Moreover, these studies do not mention the presence of two particle size modes nor distinguish between their mass spectra." Also, laser power is now listed in the experimental section.

- Page 7, Lines 26-27: Move this sentence to later in the results & discussion where Expansion 2 is discussed.

We feel that this sentence is important. We have it here to inform the reader that we will explain this result in more detail later on in the text.

- Page 7, Last paragraph: This paragraph is all repetition (summary) and could be removed as it doesn't seem necessary.

Thank you for the suggestion. We did remove many repetitive sentences in the revised

manuscript.

- Page 9, Lines 7-10: Please clarify this sentence.

This has been rewritten to read, "To examine this further, we compare in Figure S2a the MS of cloud residuals smaller than 300 nm to the reference MS of the small particle mode measured before the expansion and find them to be nearly identical. This indicates that they are bacterial fragments mixed with agar. The MS of cloud residuals that are larger than 300 nm have greater ion intensities for the organic peaks ($28CO/HNCH+$, $44CO2/CH3COH+$, $70HNCH(CH2)3+$, $85C5NH10+$) than the bacterial fragments mixed with agar, but slightly smaller ion intensities than those of the reference MS of intact bacteria, as shown in Figure S2b. These differences in ion intensities are not sufficient to eliminate the possibility that some of the larger cloud residuals could be smaller intact bacteria."

- Page 9, Line 19: Should Figure 13 be referred to here instead of Figure 6a (typo?)?

No, the correct figure is referenced.

- Page 9, Line 24: Add mention of the number concentration prior to expansion for context.

The text has been revised to read, "Figure 13c displays the welas2-measured total number of activated particles as a function of time (black), which reaches a maximum of $\sim$700 particles cm-3. Before the expansion, 1456 cm-3 particles were present in the chamber indicating that 47% of the particles activated as CCN."

- Page 10, Lines 3-4: Move "(light blue)" to after "expansion" in this sentence.

It has been moved.

- Page 10, Lines 10-12: Move sentence to the end of the previous paragraph.

These sentences have been combined with the next paragraph, not with the previous, because the previous paragraph is about size distributions and these sentences are

about the mass spectra.

- Page 11, Line 23: Please provide a reference for the hygroscopicity of intact bacteria.

A reference has been added.

- Figure 9 caption: Mention the type of bacteria used. Make sure this is included in each figure caption.

This information has been added to all figure captions.

- Figure 12: Please clarify the legend of the bottom plot.

The blue trace has been added to the legend.

- Figure 13: Please provide the sample timing for the 'cloud residuals' vs 'cloud residuals late', so that the reader can refer back to Figure 12 for context.

This has been added to the text and the figure caption.
* * *
**Fig. 1.** Response Figure 1

Fig. 2. Response Figure 2

[Figure]

**Fig. 3.** Response Figure 3

---

## Author Comment (AC2) · 15 Oct 2018

Response to Anonymous Referee #2 Thank you for your comments and suggestions. We have addressed them in the revised version of the paper. Below are our responses to your comments.

-Suski et al. present experimental data from three cloud chamber expansions at AIDA where a suspension of bacterial particles and their fragments mixed with agar were injected into the chamber. The goal appears to be to understand and contrast the role of intact bacterial cells versus bacterial fragments mixed with agar in the nucleation of cloud droplets than can then undergo immersion freezing. This is certainly a relevant

[Figure]

question to the atmospheric science community, though the presence of agar makes the results more relevant to interpreting past and future laboratory studies that use nebulized suspensions of bacterial since agar is not an atmospherically relevant particle component. In the end few new findings are really presented from these experiments, and the significance and originality of this work is rather low as a result. Perhaps the most novel aspect is the use of a cloud expansion chamber. The main finding is that intact bacterial particles make a small contribution to CCN activation and thus cloud droplets, and thus also a small contribution to immersion freezing and ice crystal production. This is certainly a worthwhile finding and it should really be made the focus of this paper, but it also requires better support from the available data.

The reviewer clearly came to the conclusion that the paper presents data which indicates that particles composed of bacterial fragments mixed with agar activated to form either droplets or ice crystals much more efficiently than intact bacteria. As the referee notes, this is an important finding. Agar or other growth media is present in all laboratory studies that are done with cultured bacteria and the present study shows that it plays an important role that needs to be considered when interpreting CCN and IN activation data. To the best of our knowledge, we present the 1st measurements of both size distributions and composition of cloud residuals of these bacteria, which made it possible to identify which particles were preferentially activated. As we note in the paper, not knowing which particles were activated led some previous studies to calculate the fractions of particles that activate as droplets and ice crystals relative to the number of intact bacteria only or to the total number of particles in the chamber. Each of these choices led to INP/CN values that do not accurately describe the activity of either particle type. Another important new finding in this study is that the relative fraction of agar in these particles gradually decreases with particle size, which affects their hygroscopicity and hence CCN and IN activity.

-I also found the paper hard to follow, and do not think the very narrative style of discussing each of the three cloud expansions to be an effective way to communicate,

analyze, and synthesize the results.

We have revised the paper to improve the flow, better illustrate the key points, and address reviewer's concerns. We also have added Table S1 that lists experimental parameters for the 3 expansions, including the CCN/CN and INP/CN ratios.

-I was not very convinced by the interpretation of these results, especially since the intact bacterial were such a small contribution to the total particle numbers to begin with. The single-particle mass spectrometer SPLAT is used to determine the chemical composition as a function of size. This analysis is hampered by the lack of significantly distinct mass spectral features that can be used to reliably distinguish the fragment+agar particles from the intact bacteria. As it is a single-particle instrument, as droplet and ice crystal activation occurs on individual particles, I found it odd that average mass spectra were presented, as opposed to trying to determine the fraction of each type of particle as a function of particle size.

It is important to keep in mind that in most previous laboratory studies of aerosolized cultivated bacteria the fraction of intact bacteria was small. Our paper is the first study that shows which of the particle types were activated as CCN and INP more efficiently when both intact bacteria and fragments mixed with agar are present at the same time. Aerosolized intact bacteria and bacterial fragments mixed with agar are composed of the same compounds, albeit present in different ratios. Therefore, it is not surprising that individual mass spectra of all particles in the chamber have the same mass spectral peaks, as shown in the new Figure 3, just with different relative intensities. All of the bacteria and bacterial fragment particles, however, contain distinct phosphorus-containing peaks that separate them from pure agar. Additionally, we demonstrate for the first time that the relative fractions of agar and bacterial fragments are changing with particle size, thereby, effecting their mass spectra. Figure 12 of the revised manuscript and Figure 3 in the response to reviewers clearly illustrate this point. As a result, distinguishing between intact bacteria and large bacteria fragments mixed with agar based on their mass spectra alone is impossible. However, a clear distinction

between these two particle types is clearly seen in the particle size distributions that show two distinct peaks, one at ∼180 nm and the other at ∼850 nm (dva). Simultaneous measurements of single particle composition and size provides the means to compare CCN and IN efficiency of two particle types. The fact that there were more particles composed of bacterial fragments mixed agar than intact bacteria does not invalidate the results. An examination of the size distributions shows that the vast majority of particles larger than 600 nm (dva) are intact bacteria. Inspection of the dva size distributions of activated particles, which does not show the distinct peak that corresponds to intact bacteria, clearly indicates that under the experimental conditions in the AIDA chamber, in which not all particles activate as cloud droplets (Table S1), the more hygroscopic smaller particle mode, composed of bacterial fragments mixed with agar, activate with significantly higher probability than intact bacteria.

-In summary, while the topic is of interest, not much new insight is presented here, and the presentation and discussion is quite unclear and hard to follow. The main singular conclusion that intact bacteria do not activate as CCN or into ice crystals needs further support. Extensive revisions are required to achieve this, and publication in ACP may not be warranted unless these major issues can be properly resolved.

We chose to present three separate experiments conducted on two bacteria to demonstrate that the data, and hence the conclusions, are perfectly reproducible. Nevertheless, we have revised the paper to improve the flow and better illustrate the key findings. We agree that given the importance of this topic the findings, which were reported here for the first time, call for follow-up studies.

-I find the narrative style of describing each expansion experimental chronologically to be an ineffective way to communicate the results. At the least the important characteristics of each expansion and how they differ from each other must be discussed. E.g. how do the aerosol concentrations and size distributions differ? How do the thermodynamic conditions of the expansion differ? It looks like expansion 1 reaches a higher supersaturation of at least 102% RHw. Stating the maximum SS/RHw reached

in each expansion is critical for understanding the CCN activation, just as stating the maximum RHi and minimum temperature reached is critical for understanding the ice crystal production.

In addition to presenting all of the RH data in the figures, we have added a table to the supplemental section (Table S1) that includes all of the expansion details. We chose this narrative style to emphasize the fact that the results of these three separate experiments, on two different bacteria, are perfectly reproducible. The differences between the thermodynamic conditions of the three expansions were minor. We agree with the reviewer that the issue of the precise value of SSw/RHw is central to most CCN and IN activation studies from which parameters for models are being calculated. This, however, is not the goal of the present study. Below, we will return to the discussion of RHw measurements. -We note that the reviewer concludes the review by stating (copied from below): "Nevertheless, the data presented here show that bacterial fragments mixed with agar preferentially activate as droplets and are the only particles observed in ice residuals." This is essentially the singular conclusion reached from this study, and it is an interesting one. I strongly suggest making this aspect the focus and providing more data and analysis that better supports this conclusion.

We agree with this conclusion. We have also revised the manuscript and added the additional data you requested.

-In describing each expansion, the authors mostly state what data is plotted in the various figures, as opposed to actually discussing, interpreting, and synthesizing the results. As written the description of the expansion experiments is not very meaningful.

We have revised the discussion to more clearly convey our interpretations of the results.

-On "hydrophobic" intact bacterial particles. So long as the particle surface is wettable these large particles will still activate under the high supersaturations reached here. It appears that RHw usually hits 101%, so 1% SS. This is why it is important to state the

maximum RHw. At high a high SS large particles with a very small hygroscopicity of kappa âĹij 0 will still activate into droplets.

The revised plots show the raw 1-sec data from tunable diode laser (TDL), which was used for in-situ measurements of the water vapor concentrations using absorption spectroscopy throughout the expansions. As discussed in detail elsewhere, (e.g. Fahey et al. (2014; www.atmos-meas-tech.net/7/3177/2014/); Hiranuma et al., 2014a (www.atmos-chem-phys.net/14/13145/2014/)) the accuracy of the measured relative humidity with respect to water (RHw) and ice (RHi) is $\pm$ 5 %. In the three expansions presented here the maximum values of the measured RHw were 97% for expansion 1, and 96% for the other two (Table S1). Given that cloud droplets were clearly formed, the values of RHw in the original plots were scaled. The high uncertainties in the RH measurements make it impossible to precisely determine SSw, which is necessary to calculate particle hygroscopicity parameters; however, this was not the goal of the present study. Again, the questions at hand are, which of the two particle types are more CCN and IN active under the same conditions, and does agar play an important role? The CCN/CN ratios listed in Table S1 indicate that not all particles in the chamber activated as cloud droplets, providing a possible explanation for why the more hygroscopic bacterial fragments mixed with agar activate, while the less hygroscopic intact bacteria do not.

-What would really help this analysis is if the /fraction/ of the two type types/size modes of particles that activate into cloud droplets and ice crystals in each expansion could be estimated. This should be possible from the data.

We calculated the fractions of particles that activated into cloud droplets and ice crystals in all three expansions and provided this information in Table S1 and in the text. Since it is impossible to distinguish between intact bacteria and large bacteria fragments mixed with agar, we cannot calculate the fraction of intact bacteria that act as CCN and INP based on MS alone. However, the size distributions of cloud residuals, shown in Figures 5, 10 and 14, exhibit no distinct peak for intact bacteria and are

dominated by smaller particles composed of bacterial fragments mixed with agar.

-It is rather misleading to say that the intact bacteria make a small contribution to cloud droplets and ice crystals considering they are a small fraction of the initial particles to begin with.

The fractions of the intact bacteria in these experiments are rather typical for laboratory studies on cultivated bacteria (e.g. Wex et al., 2015;Wolf et al., 2015;Möhler et al., 2008). If intact bacteria were, as typically assumed, more active than the bacterial fragments mixed with agar, they would be relatively enhanced in cloud residuals. Such enhancement would yield a more prominent peak at ∼850 nm in the dva size distributions, as compared to the peak observed for the particles before the expansions. We clarified the text to say that relative to the bacterial fragments mixed with agar the fraction of intact bacteria, if present in activated particles, is much smaller than that in the original sample. Wex, H., Augustin-Bauditz, S., Boose, Y., Budke, C., Curtius, J., Diehl, K., Dreyer, A., Frank, F., Hartmann, S., Hiranuma, N., Jantsch, E., Kanji, Z. A., Kiselev, A., Koop, T., Möhler, O., Niedermeier, D., Nillius, B., Rosch, M., Rose, D., Schmidt, C., Steinke, I., and Stratmann, F.: Intercomparing different devices for the investigation of ice nucleating particles using Snomax (R) as test substance, Atmos Chem Phys, 15, 1463-1485, 10.5194/acp-15-1463-2015, 2015.

Wolf, R., Slowik, J. G., Schaupp, C., Amato, P., Saathoff, H., Möhler, O., Prevot, A. S. H., and Baltensperger, U.: Characterization of ice-nucleating bacteria using on-line electron impact ionization aerosol mass spectrometry, J Mass Spectrom, 50, 662-671, 10.1002/jms.3573, 2015.

Möhler, O., Georgakopoulos, D. G., Morris, C. E., Benz, S., Ebert, V., Hunsmann, S., Saathoff, H., Schnaiter, M., and Wagner, R.: Heterogeneous ice nucleation activity of bacteria: new laboratory experiments at simulated cloud conditions, Biogeosciences, 5, 1425-1435, 2008.

-What is needed are the particle fractions, ie the CCN/CN and INP/CN ratios. This will

also make these results more useful in a quantitative manner for other researchers.

These have been added to the paper and listed in Table S1.

-As presented the results reported here cannot be used in a meaningful to for example describe the CCN or IN properties of these particles types in a model.

The goal of this manuscript is not to calculate the CCN and IN fractions to be used in a model. As we discuss in the paper, the size, number concentration, and relative fraction of the small particles containing bacterial fragments mixed with agar are affected by the solution/suspension concentrations of agar, intact bacteria, and cell fragments, how old it is, as well as how the bacteria culture was grown and prepared. Future studies need to be aware of the unavoidable presence of agar or other growth media that can greatly change the results. We cannot overemphasize that in order to use laboratory studies to generate useful parameters for atmospheric models one must make sure to eliminate artifacts that can affect CCN and IN activity. In the case of cultured bacteria, agar or other growth media is present in all laboratory studies and its effect makes it impossible to extrapolate the data to atmospheric conditions, where no agar is expected.

-Pg 1/line 30: Please provide several references for this b/g info on the role of bacteria in the atmosphere and clouds. "Murray, 2012 and references therein" is not sufficient. One suggestion: DeMott, P. J. and Prenni, A. J.: New Directions: Need for defining the numbers and sources of biological aerosols acting as ice nuclei, Atmos. Environ., 44(15), 1944–1945, doi:10.1016/j.atmosenv.2010.02.032, 2010.

References have been added.

-2/7: Contact angle would explain only the wettablilty, not hygroscopicity, of bacteria. Hygroscopicity really refers to the ability of a dissolved solute solution to take up water.

To the best of our knowledge there are no papers on the hygroscopicity of intact bacteria. The contact angle measurements were used to infer hygroscopicity in the referenced study.

-2/15: The introduction is quite spare on references to the many prior papers on the ice nucleation properties of bacteria/Pseudomonas syringae/Snomax. Some of these get mentioned much later in the paper but they also belong in the introduction, or else it appears that the authors are not familiar enough with the topic under study here. Some suggestions, but there are many more: Pandey, R., Usui, K., Livingstone, R. A., Fischer, S. A., Pfaendtner, J., Backus, E. H. G., Nagata, Y., Frohlich-Nowoisky, J., Schmu ser, L., Mauri, S., Scheel, J. F., Knopf, D. A., Po schl, U., Bonn, M. and Weidner, T.: Ice-nucleating bacteria control the order and dynamics of interfacial water, Sci. Adv., 2(4), e1501630–e1501630, doi:10.1126/sciadv.1501630, 2016. Lindow, S. E., Arny, D. C. and Upper, C. D.: Bacterial Ice Nucleation: A Factor in Frost Injury to Plants, PLANT Physiol., 70(4), 1084–1089, doi:10.1104/pp.70.4.1084, 1982. Després, V., Huffman, J. A., Burrows, S. M., Hoose, C., Safatov, A., Buryak, G., Fröhlich-Nowoisky, J., Elbert, W., Andreae, M., Pöschl, U. and Jaenicke, R.: Primary biological aerosol particles in the atmosphere: a review, Tellus B Chem. Phys. Meteorol., 64(1), 15598, doi:10.3402/tellusb.v64i0.15598, 2012. Polen, M., Lawlis, E. and Sullivan, R. C.: The unstable ice nucleation properties of Snomax[®] bacterial particles, J. Geophys. Res. Atmos., 121(19), 11,666-11,678, doi:10.1002/2016JD025251, 2016. Wex, H., Augustin-Bauditz, S., Boose, Y., Budke, C., Curtius, J., Diehl, K., Dreyer, A., Frank, F., Hartmann, S., Hiranuma, N., Jantsch, E., Kanji, Z. A., Kiselev, A., Koop, T., Möhler, O., Niedermeier, D., Nillius, B., Rösch, M., Rose, D., Schmidt, C., Steinke, I. and Stratmann, F.: Intercomparing different devices for the investigation of ice nucleating particles using Snomax[®] as test substance, Atmos. Chem. Phys., 15(3), 1463–1485, doi:10.5194/acp-15-1463-2015, 2015. Pummer, B. G., Bauer, H., Bernardi, J., Bleicher, S. and Grothe, H.: Suspendable macromolecules are responsible for ice nucleation activity of birch and conifer pollen, Atmos. Chem. Phys., 12(5), 2541–2550, doi:10.5194/acp-12-2541-2012, 2012. Turner, M. A., Arellano, F. and Kozloff, L. M.: Components of ice nucleation structures of bacteria, J. Bacteriol., 173(20), 6515–6527, 1991. Turner, M. A., Arellano, F. and Kozloff, L. M.: Three separate classes of bacterial ice nucleation structures, J. Bacteriol., 172(5), 2521–

2526, 1990. Attard, E., Yang, H., Delort, A.-M., Amato, P., Pöschl, U., Glaux, C., Koop, T. and Morris, C. E.: Effects of atmospheric conditions on ice nucleation activity of <i>Pseudomonas</i>, Atmos. Chem. Phys., 12(22), 10667–10677, doi:10.5194/acp-12-10667-2012, 2012. Hartmann, S., Augustin, S., Clauss, T., Voigtländer, J., Niedermeier, D., Wex, H. and Stratmann, F.: Immersion freezing of ice nucleating active protein complexes, Atmos. Chem. Phys. Discuss., 12(8), 21321–21353, doi:10.5194/acpd-12-21321-2012, 2012. Vali, G., Christensen, M., Fresh, R. W., Galyan, E. L., Maki, L. R. and Schnell, R. C.: Biogenic ice nuclei 2. Bacterial sources, J. Atmos. Sci., 33(8), 1565–1570, 1976.

Some of these papers were already in the introduction, but we have added the suggested references that were not there and expanded the introduction.

-2/23: Macromolecules of protein aggregates are known ice nucleants produced by bacteria. Are the macromolecules really a "solid nucleus". This definition doesn't align with the role of macromolecules as ice nucleants.

This has been revised to read, "In immersion freezing, the INP first activates as a liquid cloud droplet, when the RH over water (RHw) exceeds 100% RH, and subsequently ice forms, which means that this ice formation mechanism is tightly connected to the particle CCN activity."

-2/26: Consider using "macromolecules" instead of "nano-INP", as this is the terminology more widely used.

It has been changed to macromolecules.

-3/3: There is also evidence of biological ice nucleating macromolecules attaching to particles such as dust, and also evidence for mixed "dust-bio" particles, such as from the first author's prior work. This would also seem to motivate this study and should be discussed. Sullivan, D., Murray, B. J., Ross, J. F. and Webb, M. E.: The adsorption of fungal ice-nucleating proteins on mineral dusts: a terrestrial reservoir of atmospheric

ice-nucleating particles, Atmos. Chem. Phys., 16(12), 7879–7887, doi:10.5194/acp16-7879-2016, 2016.

This reference has been added and this point added to the introduction.

-3/27: It is a bit confusing that the stated purpose of FIN-1 is to intercompare SPMS instruments and yet that is not done here. Please explain more to better put this particular study in the context of FIN-1.

This discussion has been expanded. It now reads: "The Fifth International Ice Nucleation Workshop (FIN) was a three-part study that aimed to compare a number of single particle mass spectrometers and ice nucleation instruments. The data presented here were collected during FIN-1, which took place in November of 2014 at the AIDA cloud chamber at KIT in Karlsruhe, Germany. Ten single particle mass spectrometers from several research groups around the world were brought together for FIN-1 to simultaneously characterize the size and composition of various types of aerosol particles, including those being used in the AIDA chamber to study their activity as CCN and INPs, Various aerosol types were sampled from the AIDA cloud chamber before, during, and after expansions forming cloud particles and each of these mass spectrometers sampled using their own protocol. The results of the intercomparison will be the subject of future publication. The present manuscript presents the results of measurements, made by miniSPLAT only, during three expansions using two bacteria. "

It is important to point out that the extremely high detection efficiency and sensitivity of our SPMS, including its unmatched sensitivity to small particles, as well as its dual data acquisition mode, (Zelenyuk et al. 2015) makes it uniquely suitable to characterize both types of particles before, after, and during the expansions with high temporal resolution. Zelenyuk, A., Imre, D., Wilson, J. et al. J. Am. Soc. Mass Spectrom. (2015) 26: 257. https://doi.org/10.1007/s13361-014-1043-4

-4/14: Usually the particle density and shape are varied to arrive at a good overlap in the SMPS and APS size distributions, instead of just assuming values. Also what are

these assumed values based on? Isn't the SPLAT instrument a great way to actually measure the shape factor and density of these particles?

The values are not assumed. The effective density of the bacteria of 1.4 g cm-3 has been previously measured and presented in the paper referenced below. It was used here to achieve a good overlap of the SMPS and APS size distributions, assuming particle sphericity. We have added more clarification on this in the text. Moreover, this value is in a good agreement with effective density of 1.38 g cm-3 measured by miniSPLAT for intact pseudomonas syringae during the FIN-1 campaign.

Wex, H., Augustin-Bauditz, S., Boose, Y., Budke, C., Curtius, J., Diehl, K., Dreyer, A., Frank, F., Hartmann, S., Hiranuma, N., Jantsch, E., Kanji, Z. A., Kiselev, A., Koop, T., Möhler, O., Niedermeier, D., Nillius, B., Rosch, M., Rose, D., Schmidt, C., Steinke, I., and Stratmann, F.: Intercomparing different devices for the investigation of ice nu-cleating particles using Snomax (R) as test substance, Atmos Chem Phys, 15, 1463-1485, 10.5194/acp-15-1463-2015, 2015. Khlystov, A., Stanier, C. and Pandis, S. N.: An Algorithm for Combining Electrical Mobility and Aerodynamic Size Distributions Data when Measuring Ambient Aerosol, Aerosol Sci. Technol., 38(sup1), 229–238, doi:10.1080/02786820390229543, 2004. Beddows, D. C. S., Dall'osto, M. and Harri-son, R. M.: An Enhanced Procedure for the Merging of Atmospheric Particle Size Dis-tribution Data Measured Using Electrical Mobility and Time-of-Flight Analysers, Aerosol Sci. Technol., 44(11), 930–938, doi:10.1080/02786826.2010.502159, 2010.

-4/27: Please state the cut-size of these two CVIs.

This information has been added to the paper (Table S1 and Figures 4, 9, and 13).

-5/10: Why were these two cultures chosen? Much existing work of course on P. syringae, but what about PF CGina 01? Also unclear if these are both used in all three expansions?

Pseudomonas syringae and PF CGina are two different strains of Pseudomonas bacteria. PF CGina has been studied previously and has been shown to be an efficient INP at modestly supercooled temperatures. It has also been observed in glacier meltwater. More information about PF CGina has been added to the introduction to address this issue. Pseudomonas syringae was used in Expansion 1, while PF CGina was used in Expansions 2 and 3. The data presented in the paper indicate that the findings are consistent for both bacteria strains.

-6/15: Do you determine which particles are intact bacteria simply based on a size threshold, and if so what is it and what is it based on?

Yes, this is explained in the text. The particles in the AIDA chamber have 2 distinct size modes, with the larger size mode (dva above $\sim$700 nm) corresponding to intact bacteria.

-7/6: Qualitative terms such as "very high CCN activation efficiency" are often used. The hygroscopicity of the different aerosol types should be estimated from the maximum supersaturation observed, and the size distribution.

We have calculated the CCN/CN and have added it to the paper.

-7/14: Why would these large bacterial particle not activate? Again need to know the maximum in the RHw.

As the reviewer notes, the data presented here indicate that under these experimental conditions the large intact bacteria particles are ineffective CCN, which, in essence, implies that they are weakly hygroscopic. In contrast, smaller particles contain IN-active bacterial fragments and hygroscopic agar. Less than half of the particles in the AIDA chamber activated to form cloud droplets, thus there was competition for water vapor during the expansion or the SSw did not reach a high enough value to activate the less hygroscopic particles in the chamber. Here we are being asked to speculate as to why intact bacteria are less hygroscopic, which is beyond the scope of the present paper. We note that the interaction between water and live cells/bacteria

and the mechanisms by which they prevent water from entering into the cell are the subject of many research papers. The issue of the large uncertainties in the measured RHw in the AIDA chamber have been already discussed above.

-Have you tested that these large particles once activated into cloud droplets survive the PCVI? I agree with the other referee's comments regarding the bacterial possibly rupturing during cloud droplet activation and certainly during freezing.

The transmission efficiencies of droplets of this size have been characterized and are presented in Hiranuma et al., 2014a (www.atmos-chem-phys.net/14/13145/2014/). However, bacteria bursting in the PCVI has not been tested directly in this study, but it is not clear what the mechanism for not surviving the PCVI would be. The data do not support the idea that intact bacteria burst upon drying in the PCVI: (a) The bacteria do not all burst upon drying, as they were all dried prior to injection into the chamber and we still see intact bacteria in the chamber; (b) If they burst in the PCVI, we would expect to see a large increase in the number of particles measured by the CPC downstream of the PCVI, which was not observed; (c) Intact bacteria do not all burst upon activation or freezing because after the expansion there are still intact bacteria present in the cloud chamber. In addition, the ratio of the two particle types remain constant in the size distributions before and after the expansion, as shown in Figure 7. Therefore, we conclude that the evidence suggests that the idea of bursting intact bacteria does explain the fact that intact bacteria are not present in cloud residuals.

-Fig. 2 and other mass spectra: The two types of particles appear to have no unique ion markers, just differences in the relative ion signals. This makes separating the two particle types out based on their mass spectra rather challenging.

As we already discussed above, it is not surprising that individual mass spectra of all particles in the chamber have nearly the same mass spectral peaks, as shown in the new Figure 3, albeit with different relative intensities. Both aerosolized intact bacteria and bacteria fragments mixed with agar are composed of similar compounds

mixed at different ratios. We have found that the relative fractions of agar and bacterial fragments change with particle size, effecting their mass spectra (Figure 12 of the revised manuscript and the figure above). The distinction between the two particle types is based on simultaneous measurements of single particle composition and size. As we noted above, particles composed of bacterial fragments mixed with agar that are larger than 700 nm have the same dva and indistinguishable MS from intact bacteria. However, the dva distribution of the intact bacteria has a distinct peak at ∼850 nm, while the dva distribution of bacterial fragments mixed with agar does not have a peak in this region.

-Showing the average mass spectra is not that meaningful here. The small but important number fraction of intact bacteria will be obscured by averaging. An estimate of the number fraction of intact bacteria vs. fragments+agar in the different size modes would be really useful.

We already discussed the dependence of particle mass spectra on particle size. Moreover, if intact bacteria were more active than the fragments, they would be enhanced in cloud residuals, resulting in a more prominent larger particle mode in dva size distributions as compared to the clear peak of the intact bacteria mode detected before the expansions.

-9/14: Again, need to know what the CCN/CN fraction was at what max RHw to really evaluate the hygroscopicity of the particles. "very high CCN activation efficiency" is not quantitative.

We have provided the calculated CCN/CN fractions in the revised text and in Table S1. We discussed the issue of RHw above.

-Expansion 3 seems unique in that there is much more ice produced even though the aerosol seems similar to the other expansions, but the reason for this difference is not discussed.

Actually, during Expansion 3 the concentration of ice crystals is lower as described in the text. More ice crystal residuals were sampled and characterized during Expansion 3 because instead of the PCVI, the IS-PCVI was used to select larger cloud particles as described in the text and shown in Figure 13.

-10/14: "Nevertheless, the data presented here show that bacterial fragments mixed with agar preferentially activate as droplets and are the only particles observed in ice residuals." This is essentially the singular conclusion reached from this study, and it is an interesting one. I strongly suggest making this aspect the focus and providing more data and analysis that better supports this conclusion.

We agree, the focus of the paper was to determine which of the two particle types was more CCN and IN active. We have revised the discussion to make that more clear. We have also added the additional data you requested. It is worth noting that in addition to this main point, this manuscript demonstrates the importance of agar on cloud nucleation in laboratory studies, and presents the new finding that particle composition strongly depends on size.

[Figure]

**Fig. 1.** Response Figure 3

---

## Author Response (AR2)

**Comments to the Author:**

**Dear authors,**

**I have received reports from 2 referees, and they both suggest some additional revisions. Please consider the changes they suggest in their report and make changes in the manuscript where appropriate.**

5 **Sincerely,**

**Allan**

*Response: We would like to thank the editor and the reviewers for their careful consideration of our manuscript and for providing thoughtful and thorough comments. Below are our responses to the latest comments and questions.*

10 *Responses to Reviewer 1*

**This is a follow-up review to the revision of the Suski et al. manuscript that describes biological aerosol studies of cloud droplet and ice crystal nucleation at the AIDA chamber. In particular, the addition of the CVI cut-size on the cloud particle size distribution plots and the added detailed labeling of the mass spectra are helpful for interpretation of the residual particles. The added section headers in the results and discussion are also helpful. Please see below for**
15 **remaining comments/questions.**

**Major Comments/Questions:**

**- SPLAT Particle Size Calibration: While I do not expect this to impact the main findings in this work, in the future, the authors should take care to conduct their size calibration to encompass the entire data range, as the 70-1000**
20 **nm PSLs do not cover the full range of particles reported (up to 1.2 um), rather than relying on extrapolation. This also means that the size calibration below 70 nm is very uncertain.**

[Figure]

*Response: We agree that in general, it is beneficial to conduct size and transmission efficiency calibrations over the entire range of sizes. Figure R2_1 shows an example of typical size calibration data obtained using PSL particles with diameters between 50 nm and 1370 nm and the fit to the data ($R^2=0.9995$), using the functional form expected from FLUENT calculation. The figure clearly shows that particle size calibration derived using a slightly truncated set of particle size standards (without the smallest and the largest data points) is unlikely to introduce significant uncertainties, as the reviewer suggested. Most importantly, small uncertainty in particle sizing has no impact on this study.*

**Figure R2_1.** *A calibration curve used to convert particle time of flight to vacuum aerodynamic diameter*

*(Zelenyuk et al., 2009).*

**Figure 1 shows the small particle detection limit as a green line, but because of the scale on the axis, this cannot be determined. The detection limit should be stated in the methods section (page 6), and the miniSPLAT size distributions shown should start at this value, since the reduced particle number <0.1 um is due to scattering efficiency, rather than a lack of particles present, as shown in Figure 1 through the SMPS measurements. Further, given the reduced transmission at higher diameters, this would make the differences before and during expansion**

**size distributions even greater when this is accounted for; the authors may consider noting this in the manuscript if they haven't already, as it further supports their results.**

*Response: The 50% cut-off point of 85 nm, shown as a green line in Figures 1a and 7a, was noted in the original manuscript on pages 6-7, where we discuss the $d_m$ and $d_{va}$ size distributions presented in Figures 1. We now added this value to the*
5  *experimental section of the revised manuscript.*

*In previous publications (Zelenyuk et al., 2015;Vaden et al., 2011) we discussed the sharp decrease in detection efficiency of smaller particles in miniSPLAT, which reflects increase in particle beam divergence and decrease in light scattering signal. We noted that the drop-off is determined by the particle **true** diameter, in other words, it is not affected by particle density. As a result, the left-edge of the $d_{va}$ size distributions for particles with different densities are "shifted" from each other, as*
10  *illustrated in Figure R2_2 and discussed in (Zelenyuk et al., 2015;Vaden et al., 2011). It is, therefore, not even possible to follow the reviewer's suggestion and start the plot at the half way point, since that point depends on particle density, which is most often unknown.*

*Consequently, we always plot the **measured** $d_{va}$ size distributions and provide an explanation for the shape of the measured data on the small particle side. The precise 50% cut-off point of miniSPLAT, which can vary between 79 nm and 85 nm, due*

[Figure]

**Figure R2_2.** *(a) log-normal particle size distribution (red) and the corresponding calculated $d_{va}$ size distribution. The latter was calculated for spherical particles with density of $\rho$= 1.0 g cm$^{-3}$, using a 50% cut-off diameter of 109 nm for the older SPLAT II instrument (Vaden et al., 2011); (b) Calculated $d_{va}$ distributions of particles with the densities indicated in the legend. Also shown are the densities estimated using the half-way points of the left-edge of the $d_{va}$ distributions method described in Vaden et al. (2011).*

15  *instrument alignment and the optics quality, is measured for each miniSPLAT deployment, using spherical particles with known or precisely measured (±0.5%) density (Zelenyuk et al., 2015;Vaden et al., 2011).*

*It is worth pointing out that it is possible to use the measured $d_{va}$ size distribution of poly-dispersed particles with known density to determine the exact shape of the detection efficiency of small particles (Vaden et al., 2011). We also demonstrated that the 50% cut-off point of the miniSPLAT-measured $d_{va}$ size distributions can be used to determine particle density, as*
20  *illustrated in Figure R2_2.*

**-        Page 7, Line 16 and elsewhere: Could m/z 30 be NO+?**

*Response: Yes, as labeled on the mass spectra presented in the figures, the possible assignment for peak at m/z=30 is $H_2NCH_2^+$ and $NO^+$. We corrected the text.*

**Technical Comments/Questions:**

**-        Page 2, Lines 20-22: Please provide a reference for this sentence.**

*Response: The reference was added.*

**-        Page 6, Line 8: Fix typo – "um" should be "nm".**

*Response: The typo was corrected.*

**-        Page 6, Lines 9-10: The authors state that "The transmission efficiency through the aerodynamic lens is expected to change by only a few percent with the observed pressure changes during the expansion, as shown in Liu et al. (2007)." However, in Figure 2 of the author response, the change in transmission efficiency at >500 nm is nearly 20%, which is much more than "only a few percent". Please revise this sentence appropriately.**

*Response: The sentence was revised.*

**-        Page 7, Line 16: Fix typo – 18NH3+ should be 18NH4+. Also, revise sentence so as to not list ammonium as an organic peak.**

*Response: The sentence was revised.*

**-        Page 7, Lines 25-29: I suggest removing these sentences from the manuscript, as they are not necessary, and in comparing the spectra shown herein with these papers, I think this is subjective to interpretation without a rigorous comparison. Also, it neglects that there have been many other biological SPMS papers that have looked at the intensities of organic ion peaks with laser intensity (e.g. Steele et al., 2003, Analytical Chem.), for example (which would be more appropriate to discuss, if the authors wanted to keep a comment on the organic ion intensities).**

*Response: The main goal here was to compare our MS of intact Pseudomonas syringae bacteria to the MS of the **same** bacteria acquired by other single particle mass spectrometers. As we discuss in the manuscript, all the negative ion MS are virtually the same, while our positive ion MS exhibit significantly higher intensities of organic ion peaks.*

*Note that the Steele et al. (2003) paper does not present a study of the intensity of organic ion peaks. All MS peaks in the positive ion MS of Bacillus subtilis var. niger, presented in (Steele et al., 2003), contain Ca. The only exception is an unidentified peak at m/z=74.3, the relative intensity of which is decreasing with increased laser fluence. In addition, several other biological SPMS papers that were used by us to aid in ion identifications were cited in the original version of our manuscript (e.g. (Pratt et al., 2009;Zawadowicz et al., 2017;Wolf et al., 2015;Fergenson et al., 2004;Srivastava et al., 2005;Czerwieniec et al., 2005).*

**-        Page 8, Line 27: Fix typo – 18NH3+ should be 18NH4+.**

*Response: The typo was corrected.*

**Figure 3: Since this figure only include m/z labels as numbers, it would be helpful for the Figure 2 labels to include numbers, in addition to the formulas. Also, it would be helpful to add formula labels to the positive ions as you are able to, and use consistent labeling between the positive and negative ion mass spectra. Please also label in the caption if these are average mass spectra. For plots a & c, it would be helpful to show only the scaled (e.g. x10) portion of the mass spectrum for a given m/z range, rather than the original plus this scaled part, as the spectra are currently very busy and showing the peaks twice makes it harder to read.**

*Response: We modified Figure 3 to show only the scaled portions of the mass spectra and revised the figure caption as suggested by the reviewer. We did, however, refrained from extending the peak labels in these crowded figures. The observed MS peaks, their molecular identification, and m/z are provided in the text.*

**The authors have largely addressed the concerns raised during review, and have increased the clarity of the manuscript, while also adding new results and figures. I still find the scientific significance of this work to be rather limited. From the author's response I now understand the intent of these experiments and analysis in the context of better understanding similar prior experiments using nebulized suspensions of bacteria with growth media. The manuscript would really benefit if this motivation was clearly stated early in the paper to give the study focus and a clear purpose. A sentence in the Abstract and a paragraph in the Introduction explaining the purpose of these experiments, in context of prior laboratory experiments, would add a lot of value. Right now this discussion only comes at the very end in the Conclusions. The authors already provided a clear explanation of this in their response to my review, but regrettably this did not get added to the revised manuscript. Thus it would be easy for them to add to the actual paper.**

*Response: We revised the manuscript to clarify this point.*

**It would be valuable to put bounds on the kappa hygroscopicity parameter of the intact bacterial particles since they did not CCN activate. This can be done since you know their size distribution, and have an estimate of the max RHw reached in the chamber. From this you can put an upper limit on what their kappa value must have been such that they did not activate due to low effective hygroscopicity.**

*Response: To get a rough estimate for kappa (k), or the hygroscopicity parameter, of the intact bacteria we used plots of critical supersaturation vs. dry dimeter for particles of different kappa, presented by Petters and Kreidenweis (Figure 1 in (Petters and Kreidenweis, 2007)). Based on the size distribution of particles in the AIDA chamber and the number of activated droplets, we find that particles composed of agar mixed with bacterial fragments as small as 80 nm activated as droplets. If we use kappa of 0.3 (between that of sucrose and ammonium sulfate), we can estimate that the supersaturation must have been around 0.3%. The fact that the ~800 nm intact bacteria did not activate at this supersaturation yields kappa≈0 for intact bacteria, which remains true even if we use kappa=1 for agar/bacteria. In other words, intact bacteria are expected to activate under wetting conditions. This is consistent with some previously published experimental results, which indicate that intact bacteria are non-hygroscopic. It is also consistent with the fact that intact live bacteria do not dissolve in water, or uptake it.*

*We agree that it would be nice to be able to put a precise number on hygroscopicity and CCN activation of intact bacteria, however, the fact is that in the experiments presented here the RHw was not sufficiently well determined (±5%RH) to allow for precise determination of kappa for these particles and its inclusion in the manuscript.*

[revised manuscript text omitted]